# Single-nucleus RNA-sequencing of autosomal dominant Alzheimer disease and risk variant carriers

Logan Brase [1,2,3], Shih-Feng You[1,2,3], Ricardo D'Oliveira Albanus [1,2,3], Jorge L. Del-Aguila [4], Yaoyi Dai [5], Brenna C. Novotny [1,2,3], Carolina Soriano-Tarraga[1,2,3], Taitea Dykstra [6,7], Maria Victoria Fernandez [1,2,3], John P. Budde [1,2,3], Kristy Bergmann[1,2,3], John C. Morris[2,8,9], Randall J. Bateman [2,8,9], Richard J. Perrin [2,6,8,9], Eric McDade [1], Chengjie Xiong[8,10], Alison M. Goate [11], Martin Farlow[12], Dominantly Inherited Alzheimer Network (DIAN)*, Greg T. Sutherland [13], Jonathan Kipnis [6,7], Celeste M. Karch [1,2,3,15], Bruno A. Benitez[14,15] & Oscar Harari [1,2,3,15] ✉

Genetic studies of Alzheimer disease (AD) have prioritized variants in genes related to the amyloid cascade, lipid metabolism, and neuroimmune modulation. However, the cell-specific effect of variants in these genes is not fully understood. Here, we perform single-nucleus RNA-sequencing (snRNA-seq) on nearly 300,000 nuclei from the parietal cortex of AD autosomal dominant (*APP* and *PSEN1*) and risk-modifying variant (*APOE, TREM2* and *MS4A*) carriers. Within individual cell types, we capture genes commonly dysregulated across variant groups. However, specific transcriptional states are more prevalent within variant carriers. *TREM2* oligodendrocytes show a dysregulated autophagy-lysosomal pathway, *MS4A* microglia have dysregulated complement cascade genes, and *APOEε*4 inhibitory neurons display signs of ferroptosis. All cell types have enriched states in autosomal dominant carriers. We leverage differential expression and single-nucleus ATAC-seq to map GWAS signals to effector cell types including the *NCK2* signal to neurons in addition to the initially proposed microglia. Overall, our results provide insights into the transcriptional diversity resulting from AD genetic architecture and cellular heterogeneity. The data can be explored on the online browser (http://web.hararilab.org/SNARE/).

Alzheimer's disease (AD), the most common cause of dementia, is characterized by amyloid (Aβ) plaques and neurofibrillary tangles (NFTs, Tau deposits) in the brain, accompanied by neuroinflammation, myelination changes, synaptic dysfunction and loss, gliosis, and neuronal death[1,2]. Genetic studies have successfully identified multiple genes and pathways associated with AD pathogenesis, including rare mutations in *amyloid precursor protein* (*APP*)[3,4], *presenilin 1* (*PSEN1*)[5],

and *presenilin 2* (*PSEN2*)[6] that cause autosomal dominant AD (ADAD), mainly by driving aberrant Aβ production in support of the amyloid cascade hypothesis[7]. The strongest risk factor for sporadic AD (sAD), *apolipoprotein E* (*APOE*) ε4, implicates cholesterol metabolism and Aβ clearance mediated by astrocytes and microglia[8]. Integration of epigenetics and genome-wide association studies (GWAS) signals have involved immune system dysfunction in AD[9]. Low-frequency variants

A full list of affiliations appears at the end of the paper. *A list of authors and their affiliations appears at the end of the paper. ✉e-mail: harario@wustl.edu

in *triggering receptors expressed on myeloid cells 2* (*TREM2*), a gene that drives microglia activation, increase AD risk[10]. Recent GWAS have identified additional common genetic risk factors for AD, including rs1582763, an intergenic variant in the *MS4A* locus that confers resilience to AD and is associated with higher cerebrospinal fluid (CSF) soluble TREM2 levels[11–14]. However, linkage and association studies do not reveal downstream transcriptional ramifications nor the cell types these variants influence.

Single-nucleus RNA-sequencing (snRNA-seq) has emerged as a powerful approach to interrogating the underlying transcriptional landscape of the cellularly-complex human brain. SnRNA-seq has been used to study AD in different cohorts and brain regions, including the entorhinal cortex[15], temporal neocortex[15,16], prefrontal cortex[17,18], and the dorsolateral prefrontal cortex (DLPFC)[16,19,20]. However, no single-cell study has analyzed genetic high-risk variants and mutations in the AD parietal cortex. A systematic and unbiased survey of cell type-specific gene expression across this region will help identify transcriptional states associated with the high Aβ plaque and tangle burden but relatively low atrophy in the early stages of AD, which are characteristic of the parietal cortex[21].

In this study, we selected participants from the Dominantly Inherited Alzheimer Network (DIAN) and Charles F. And Joanne Knight Alzheimer Disease Research Center (Knight ADRC) biobank to enrich our cohort with genetically defined AD patients carrying pathogenic, risk, and resilience genetic variants to leverage the naturally occurring perturbations of these genes and elucidate their role in AD pathogenesis. Specifically, we performed snRNA-seq on the parietal cortex (Brodmann areas 7 and 39). After stringent filtering, we obtained 294,114 high-quality nuclei from 67 individuals. We iteratively extracted nuclei by cell type (digital sorting), subclustered the nuclei into cell type transcriptional states (cell states), and identified differentially expressed genes (DEGs) by cell states. We identified differences in cell-type composition associated with ADAD, *APOEε4*, *TREM2*, and *MS4A* carriers and identified DEGs between the genetic groups and control samples. Finally, we created an online browser (http://web.hararilab.org/SNARE/) for convenient, unrestricted access to the snRNA-seq expression data.

We independently replicated the genetically driven transcriptional and proportional profiles identified in the parietal cortex using snRNA-seq data of the DLPFC from the ROSMAP cohort[20], snRNA-/snATAC-seq data of the prefrontal cortex from UCI MIND's ADRC[17], and single-cell RNAseq of 5xFAD mouse microglia[22] (see details in Methods).

Our findings highlight the power of leveraging genetic and single-cell molecular data to understand the heterogeneity of pathways, biological processes, and cell types mediating AD genetic risk factors.

## Results

### A single-nucleus atlas captures the transcriptional diversity among sporadic and autosomal dominant AD

Detailed clinical data, postmortem neuropathological data, and genetic characterization of the discovery cohort are described in Table 1. We generated whole transcriptomes of the parietal cortices at single-cell resolution using 10x Genomics Next GEM technology. After data cleaning and quality control (QC; see Methods), we retained 67 samples and obtained molecular data for 16 carriers of pathogenic mutations in *APP* and *PSEN1* (autosomal dominant AD; ADAD), 31 sAD non-carriers of risk-modifying variants, three individuals who matched AD-neuropathological criteria but without clinical cognitive impairment at age-of-death (presymptomatic), eight individuals who matched non-AD neurodegenerative pathologies criteria (other), and nine individuals who exhibited neither neurodegenerative pathology nor evidence of dementia (control) (Table 1). Within the cohort, 41 samples carried the minor allele (A) for rs1582763, an SNP in the *MS4A* gene cluster[11,13,14], 24 samples carried the *APOEε4* allele, and 19 samples

**Table 1 | Demographic characteristics of samples**

| Samples | Control | ADAD | sAD | Presym | Other |
|---|---|---|---|---|---|
| Total | 9 | 16 | 31 | 3 | 8 |
| MS4A (AG %)* | 55.6 | 46.7 | 45.2 | 33.3 | 12.5 |
| TREM2@ | - | - | 15 | - | 4 |
| PSEN1 | - | 13 | - | - | - |
| APP | - | 3 | - | - | - |
| Braak Aβ (O/A,B/C) | 2/7/0 | 0/0/16 | 0/0/31 | 0/0/3 | 2/5/1 |
| Braak Tau (NA/I-III/IV-VI) | 0/9/0 | 3/0/13 | 4/2/25 | 0/0/3 | 0/6/2 |
| Sex (XY)% | 33.3 | 56.3 | 45.2 | 33.3 | 50.0 |
| AOD (mean, sd)y | 90.1(9.6) | 51.0(6.9) | 81.5(6.4) | 77.3(15.3) | 88.8(6.1) |
| APOEε4+%$ | 11.1 | 25.0 | 54.8 | 33.3 | 12.5 |
| PMI (mean, sd)h | 10.9(5.5) | 14.2(7.7) | 11.9(6.3) | 12.4(1.9) | 11.3(9.1) |

Other: (1:Dementia with Lewy bodies, 4:Argyrophilic grain disease, 1:Tramatic encephalopathy, 1:Neurofibrillary tangle-predominant AD, 1:Cerebrovascular disease).
*ADAD* autosomal dominant Alzheimer's disease, *sAD* sporadic Alzheimer's disease, *Presym* presymptomatic, *PMI* postmortem interval.
*MS4A is referring to SNP rs1582763 (**GG:25, AG:28, and AA:13**).
@Two African descent and one Asian descent (the p.H157Y is European descent).
$The total number of *APOE* ε4+ were 24 (*APOE* genotypes: **23**:4, **24**:2, **33**:39, **34**:19, **44**:3).

(sAD:15, other:4) carried *TREM2* risk variants (Table 1, Fig. 1a, and Supplementary Dataset 1).

After data cleaning and QC (Table 2), we retained and clustered 294,114 nuclei (Supplementary Dataset 2), identifying 15 clusters covering major brain cell types (Fig. 1b, c). These clusters were annotated based on the expression of well-known cell-type markers[23] (Fig. 1c and Supplementary Dataset 3). Clusters 0, 1, 2, and 9 were identified as oligodendrocytes, clusters 3, 8, 10, 11, and 13 as neurons, and clusters 4, 6, and 14 as astrocytes. Microglia, oligodendrocyte precursor cells (OPCs), and endothelial cells were identified in clusters 7, 5, and 12, respectively (Fig. 1b, c). We confirmed that all samples and batches were fairly distributed among the different clusters (Entropy−See Methods, Supplementary Figs. 1, 2, Supplementary Datasets 4, 5). The distribution of nuclei by clusters (Supplementary Dataset 2) showed that glial cells accounted for 80.7% of the nuclei, whereas neuronal nuclei accounted for 18.2% (the remaining ~1% were endothelial cells). These proportions are consistent with histological and bioinformatic cell-type composition[24]. OPCs (generalized linear regression β = 0.15, $p = 2.99 \times 10^{-2}$) and endothelial cells (β = 0.07; $p = 1.25 \times 10^{-2}$) showed elevated proportions in ADAD participants compared to controls. All other cell types were similarly represented across AD statuses (Supplementary Dataset 6a).

After subclustering each cell type, we identified five to nine subclusters or cell-type transcription states (cell states) within each cell type (Supplementary Dataset 2). These cell states were generally well represented among samples and batches (Supplementary Fig. 1 and Supplementary Dataset 5). Cell states flagged for review (<half the total entropy possible) were later identified as either having few nuclei or being enriched for carriers of genetic factors (i.e., ADAD, *MS4A*, or *TREM2*). Neuronal cell states were then categorized as excitatory (EN) or inhibitory (IN; See Methods). We capture an average of 2696 upregulated genes (out of an average of 16,223 genes tested) that passed multiple testing correction in each cell state compared to all other cell states in the same cell type (Supplementary Datasets 7, 8, 9), which reveals a rich transcriptional diversity within all brain cell types. As expected, due to statistical power, there was a positive correlation between the number of detectable DEGs and the number of nuclei

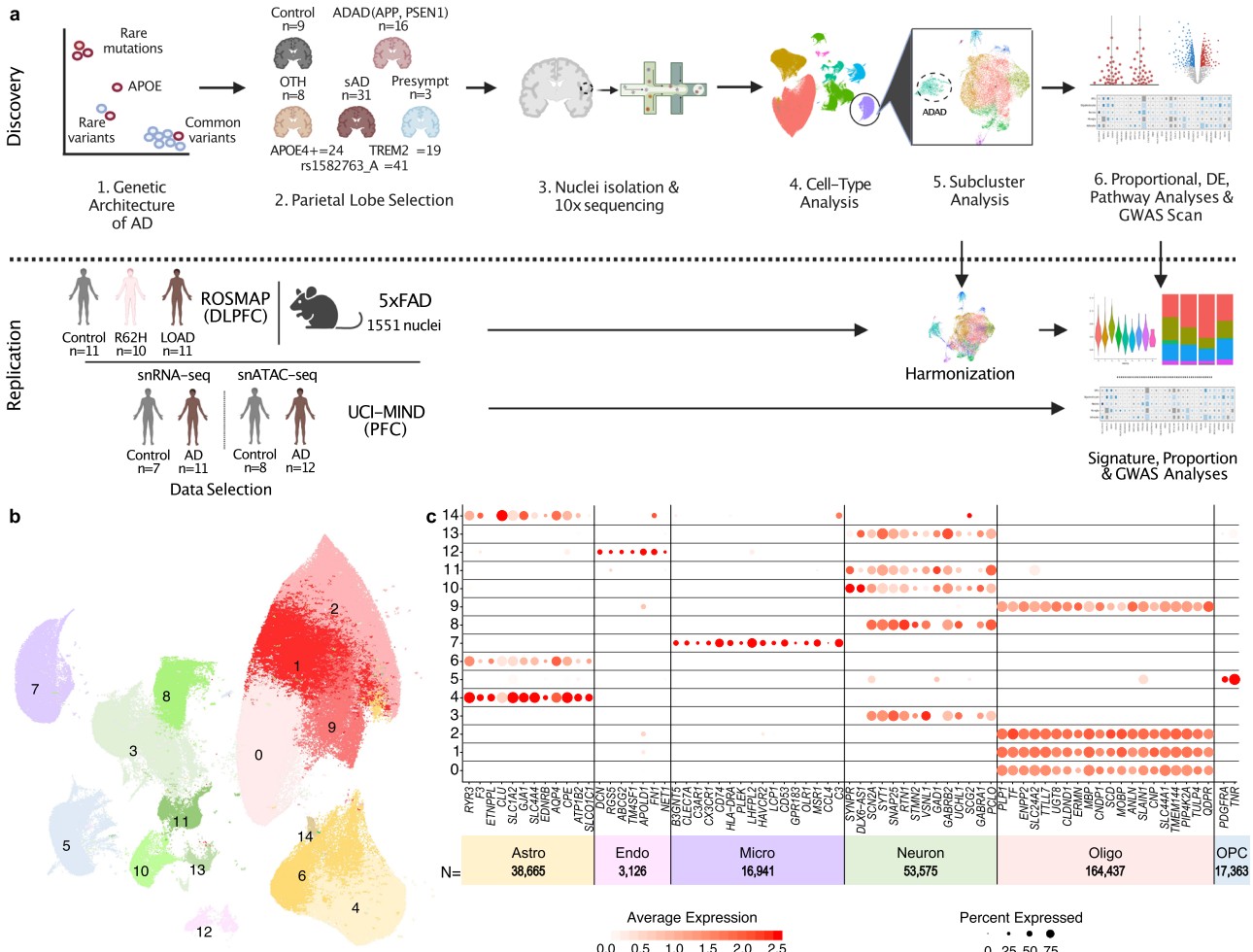

**Fig. 1 | SnRNA-seq distinguishes major cell types using 67 human brains.**
**a** Diagram of the study design. **b** UMAP plot showing 15 distinguished clusters, 0–14, with 294,114 total cells. **c** DotPlot depicting expression of cell-type-specific markers genes to identify each cluster in **b**. Source data are provided as a Source Data file.

### Table 2 | Quality control steps

| Performed on | Individual samples | | | | | | | Cell types |
|---|---|---|---|---|---|---|---|---|
| filter applied | Cell ranger | Barcode-rank Distribution | Distrib UMI | Distrib genes | % mit. | >10 cells* | Double finder | Garnett |
| Total #nuclei | 1,102,459 | 566,486 | 423,937 | 414,303 | 380,625 | 380,625 | 346,264 | 294,114 |
| Mean #genes | 3154 | 2418 | 2242 | 2228 | 2234 | 2234 | 2221 | 2182 |
| Mean #UMI | 10,279 | 5646 | 4969 | 4899 | 4897 | 4897 | 4888 | 4725 |
| Avg %mitochondria | 2.35 | 2.34 | 1.98 | 1.94 | 1.22 | 1.22 | 1.21 | 1.21 |

*mit* mitochondria.
# number
*Genes were removed if they were not expressed in at least ten nuclei.

in the cell type. A notable exception: most likely due to the neuronal differences across cortical layers[25], excitatory and inhibitory neurons exhibited the highest average number of DEGs despite not having the greatest nuclei abundance (Supplementary Dataset 7). We identified a comparatively small number of endothelial nuclei, which were not further subclustered (Supplementary Dataset 2).

**Shared transcriptomic profiles across AD groups and cell types**
We leveraged the snRNA-seq data to compare the cell-type-specific gene expression patterns of the sAD, *TREM2*, and ADAD groups compared to controls (Supplementary Dataset 9). ADAD donors have more underexpressed than overexpressed genes for each cell type.

Conversely, *TREM2* samples have more overexpressed than underexpressed genes (Supplementary Datasets 10, 11). Astrocytes, excitatory neurons, and OPCs show a trend for transcriptional underexpression across groups; in contrast, microglia and oligodendrocytes show overexpression (Fig. 2a). This suggests that, in general, astrocytes, excitatory neurons, and OPCs lose functionality overall while microglia and oligodendrocytes increase in functionality in AD. A notable exception is the *TREM2* group's OPCs, which generally show overexpression compared to controls.

We then compared whether genes were coincidentally differentially expressed across AD groups (Fig. 2b and Supplementary Fig. 3). In many instances, sAD and ADAD have the same direction of effect on

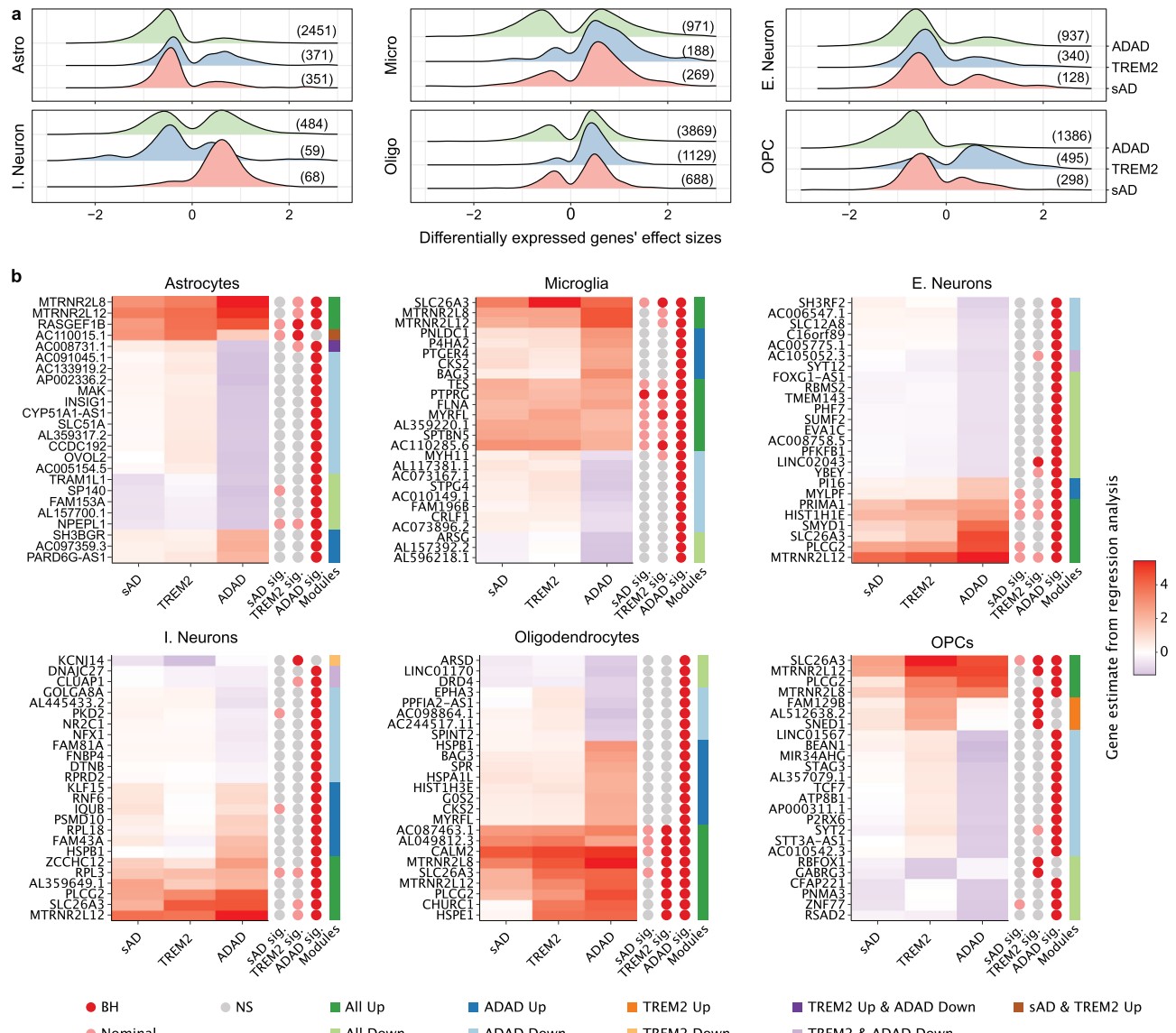

**Fig. 2 | Divergent and coincident expression patterns across ADAD, *TREM2*, and sAD by cell type. a** Ridge plots showing the distribution of gene estimates extracted from the linear mixed models comparing control nuclei to ADAD, *TREM2*, and sAD nuclei within cell types. Only DEGs that passed multiple testing correction in at least one genetic group were considered. For inclusion, the gene must also have been nominally significant ($p < 0.05$) in the genetic group. The total number of genes is shown on the right tail. **b** Heatmaps of gene estimates from the same models emphasize the divergent and congruent expression patterns across genetic groups. The largest 500 estimates were selected per cell type and used to create full heatmaps (found in Supplement). "Modules" were manually created based on expression patterns and dendrogram groupings. The top 10% of genes from each module were extracted (order preserved) to produce the above heatmaps. "sAD sig.", "*TREM2* sig.", and "ADAD sig." depict the significance status of each gene for that group. "BH": Benjamini−Hochberg $p < 0.05$, "Nominal": $p < 0.05$, "NS": not significant. Source data are provided as a Source Data file.

differential gene expression. However, the effect tends to be stronger in ADAD (Supplementary Fig. 4). We also identified group-specific gene patterns of interest. Within ADAD astrocytes, there was over-expression of genes *SLC7A5*, *LRP2*, and *SLC7A1* related to "transport across blood−brain barrier" (Benjamini−Hochberg $p = 4.09 \times 10^{-2}$). Also within ADAD astrocytes, was the overexpression of *ITGA10*, *SER-PINF2*, *P3H2*, *PLOD3*, *PLOD1*, and *ADAMTS12* related to "extracellular matrix organization" (Benjamini−Hochberg $p = 1.35 \times 10^{-2}$) which is concordant with other evidence implicating astrocytes in matrisome perturbation[26]. Across AD groups in excitatory neurons, we identified overexpression of *EHD1*, *DGAT2*, *LRP5*, and *LDLRAP1* relating to 'cholesterol homeostasis' (Benjamini−Hochberg $p = 3.99 \times 10^{-2}$). Unique to *TREM2* OPCs, we captured overexpression of genes *PDGFRB*, *PFKP*, and *PDK1* relating to "Central carbon metabolism" (Benjamini−Hochberg $p = 3.94 \times 10^{-2}$, Supplementary Dataset 12).

We found distinctive patterns of shared dysregulated genes across cell types (Supplementary Fig. 5 and Supplementary Datasets 13, 14). All cell types showed dysregulation of "lysosome" genes *HGSNAT* and *ARSG* (Benjamini−Hochberg $p = 2.86 \times 10^{-2}$) in ADAD samples compared to controls. In ADAD, compared to sAD samples, genes related to "vesicle-mediated transport" (*CCZ1B*), "Golgi maintenance" (*GOLGA8B*), and "vesical docking" (*RABEPK*) were dysregulated in all cell types.

In astrocytes, *SULT1A2* and *SQSTM1* were overexpressed in sAD, *TREM2*, and ADAD groups compared to controls (Supplementary Fig. 5 and Supplementary Datasets 15, 16). *SULT1A2* catalyzes the sulfate conjugation of hormones, neurotransmitters, drugs, and xenobiotics. The *SQSTM1* gene encodes p62, a protein involved in the signaling for multiple pathways relating to proteasomes, autophagy, oxidative stress, inflammation, and immune response[27]. It has been proposed as

a therapeutic target to facilitate amyloid-β removal by autophagy activation[28].

In microglia, the genes *FLT1* and *PTPRG* were overexpressed in sAD, *TREM2*, and ADAD, whereas *FARP1* was overexpressed in sAD, ADAD, and *APOEε4+* (see Methods) samples. *FLT1* mediates microglial chemotactic inflammatory responses, which contribute to pathological conditions in the AD brain[29]. *PTPRG* is a tyrosine phosphatase, and a family-based GWAS hit for AD risk[30]. *FARP1* is involved with synapse formation, and the retention of a specific intron within this gene is associated with AD in mouse brains[31].

The gene *SNTG*, which is associated with the age of onset in *PSEN1* p.E280A carriers[32], was dysregulated in excitatory neurons of ADAD and *APOEε4+* samples. In oligodendrocytes, *LPL* and *VWA3B* are both overexpressed in *TREM2* and ADAD compared to controls and ADAD compared to sAD. *LPL*, lipoprotein lipase, is increased in AD brains and associated with AD progression[33]. Mutations in *VWA3B* cause a recessive form of Spinocerebellar Ataxia[34].

In OPCs, *CNTNAP2* is overexpressed in the ADAD and *APOEε4+* groups and underexpressed in the *TREM2* group. *PTPN13* is overexpressed in the *TREM2* and ADAD groups compared to controls and in ADAD compared to the sAD group. *CNTNAP2* is required for jap junction formation. It is an AD GWAS hit linked to neurodevelopmental disorders, including Tourette syndrome, autism, schizophrenia, epilepsy, OCD, and ADHD[35]. *PTPN13* is a tyrosine phosphatase involved in tau tyrosine phosphorylation[36]. These genes highlight core biological functions dysregulated across multiple genetic backgrounds.

## Pleiotropic effects of *APP* and *PSEN1* mutations across brain cell types

Strikingly, every cell type we analyzed exhibited cell states enriched within ADAD carriers (Fig. 3a–e, Supplementary Fig. 6, and Supplementary Datasets 6b, 9). Two astrocyte cell states were specific to ADAD samples. One in particular, Astro.4 (Astro-DAA; generalized linear regression β = 0.15, $p = 4.39 \times 10^{-2}$), had increased expression of *OSMR* (linear mixed effects regression log2FC = 1.48, Benjamini–Hochberg $p = 4.37 \times 10^{-4}$), *VIM* (log2FC = 1.80, Benjamini–Hochberg $p = 8.92 \times 10^{-13}$), and *CTSB* (log2FC = 1.56, Benjamini–Hochberg $p = 4.03 \times 10^{-5}$) compared to the other astrocyte cell states, recapitulating the expression profile of the disease-associated astrocytes (DAA) identified in the 5xFAD mouse model[37] (Supplementary Fig. 7f). Astro-DAA's overexpressed genes are associated with "cytoplasmic translation" (Benjamini–Hochberg $p = 9.06 \times 10^{-22}$) and "cytokine-mediated signaling" (Benjamini–Hochberg $p = 3.05 \times 10^{-15}$) (Fig. 3f and Supplementary Dataset 17b).

Mic.4 (Mic-stress) was a prominent cell state enriched in ADAD samples (general linear regression β = 0.40, $p = 2.50 \times 10^{-3}$; Supplementary Datasets 6b, 9), capturing nuclei from 11 *PSEN1* carriers and one *TREM2* p.R136W carrier (Fig. 3b and Supplementary Fig. 7). Mic-stress's DEGs did not overlap with established microglia signatures (hypergeometric), including "disease-associated microglia" (DAM)[38], "microglial neurodegenerative" (MGnD)[39], and "human AD microglia" (HAM)[40] (Supplementary Dataset 18). Interestingly, Mic-stress showed a significant increase in the expression of *MECP2* (linear mixed effects regression β = 0.67, Benjamini–Hochberg $p = 3.20 \times 10^{-10}$, Supplementary Fig. 7g), which has previously shown differential expression in AD brains and, when knocked down in microglia, caused NMDA receptor-dependent excitotoxic neuronal cell death in a mouse model of Rett syndrome[41]. Pathway analysis revealed that the 491 upregulated genes in Mic-stress were associated with the regulation of "cellular response to stress" (Benjamini–Hochberg $p = 3.17 \times 10^{-3}$) and "receptor-mediated endocytosis" (Benjamini–Hochberg $p = 1.74 \times 10^{-2}$) (Fig. 3g and Supplementary Dataset 17c). Furthermore, the Mic-stress upregulated gene signature was replicated (upregulation signature score generalized linear regression $p$ range $1.92 \times 10^{-374}$ to $2.45 \times 10^{-28}$; Fig. 3h) in the

cortex of seven-month-old 5xFAD mice (high amyloid plaque load at this age).

Oligo.3 (Oligo-spliceosome) was also significantly associated with ADAD samples (Fig. 3e, generalized linear regression β = 0.63, $p = 1.48 \times 10^{-6}$; Supplementary Datasets 6b, 9). Pathway analysis of the Oligo-spliceosome upregulated genes was enriched in genes related to 'mRNA splicing, via spliceosome' (Benjamini–Hochberg $p = 1.42 \times 10^{-41}$, Fig. 3g) mainly from the family of heterogeneous nuclear ribonucleoproteins (HNRNP), including *HNRNPA1, HNRNPA2B1, HNRNPA3, HNRNPC, HNRNPD, HNRNPH3, HNRNPK, HNRNPM,* and *HNRNPU*, which have previously been linked to late-onset AD (Supplementary Fig. 7h and Supplementary Dataset 17h)[42]. The AD risk genes, *PICALM, CLU, APP,* and *MAP1B*, which have intronic excision levels correlated with the expression of HNRNP genes[42], were also overexpressed in Oligo-spliceosome (Supplementary Fig. 7h). This suggests that *PICALM, CLU, APP,* and *MAP1B* could be alternatively spliced through HNRNP splicing repression within oligodendrocytes. HNRNP genes also play a role in amyotrophic lateral sclerosis (ALS) and frontotemporal dementia (FTD)[43] and promote the translation of *APP*[44].

Additional cell states in oligodendrocytes (Oligo.1), OPCs (states 4, 5, 6), and neurons (EN.3, IN.2) were enriched within ADAD carriers (Fig. 3c–f and Supplementary Datasets 6b, 9).

## *TREM2* variants modulate microglial and oligodendrocytic transcription states

The *TREM2* AD-risk variants associated with reduced cellular activation (p.R47H, p.R62H, and p.H157Y)[20,45] were associated with Mic.2 (Mic-reduced; Fig. 4a; generalized linear regression β = 0.23, $p = 3.29 \times 10^{-2}$; Supplementary Datasets 6c, 9). Mic-reduced showed high expression of resting-state-microglia marker genes (*TMEM119, P2RY13, MED12L,* and *SELPLG*) and minimal elevation of activated (*ABCA1, C5AR1, TNFAIP3,* and *CD83*) marker genes compared to the Mic-resting (Mic.0) and Mic-activated (Mic.1) cell states (Fig. 4b, Supplementary Fig. 8, and Supplementary Dataset 18). We analyzed microglia from 32 ROSMAP snRNA-seq samples (DLPFC) that included 11 *TREM2* p.R62H carriers (See Methods, Synapse ID: syn21125841)[20]. We found that 10.2% of the ROSMAP microglia recapitulated the Mic-reduced transcriptional state (signature score generalized linear regression $p$ range $1.04 \times 10^{-165}$ to $4.56 \times 10^{-3}$; Fig. 4c, d and Supplementary Figs. 9, 10). Carriers of *TREM2* p.R62H had a higher proportion of their microglia in Mic-reduced than did non-carriers (Fig. 4a, Discovery: generalized linear regression β = 0.20, $p = 2.58 \times 10^{-2}$; Meta-analysis: $p = 2.26 \times 10^{-2}$).

In the discovery cohort, these *TREM2* risk variant carriers were also enriched for Oligo.5 (Oligo-*TFEB*; generalized linear regression β = 0.13, $p = 4.66 \times 10^{-2}$; Fig. 4e and Supplementary Dataset 6c), which exhibited upregulation of 1124 genes including *TFEB* (linear mixed effects regression Log2FC = 0.15; Benjamini–Hochberg $p = 8.69 \times 10^{-6}$; Supplementary Dataset 8) compared to other oligodendrocyte cell states (Fig. 3d). Altered *TFEB* expression may be driven by the interaction of *TREM2* with *mTOR*, which is upstream of *TFEB*[46,47]. *TFEB*, a central regulator of lysosomal biogenesis and autophagy[48,49], represses myelination at different developmental stages[50] and dysregulated *TFEB* signaling has been implicated in multiple neurodegenerative diseases[48,51]. Oligo-*TFEB* also had increased proportions among sAD samples compared to controls (Supplementary Dataset 6c). Within the ROSMAP cohort, 7.1% of oligodendrocytes exhibited this transcriptional signature (upregulation signature score of Oligo-*TFEB* generalized linear regression $p$ range $1.22 \times 10^{-483}$ to $3.56 \times 10^{-2}$; Fig. 4g; Supplementary Figs. 9, 11) with an increased proportion in *TREM2* p.R62H carriers (Discovery: generalized linear regression β = 0.13, $p = 2.48 \times 10^{-2}$; Meta-analysis: $p = 6.11 \times 10^{-3}$; Fig. 4e). Furthermore, our analysis of the expression-derived gene regulatory network (GRN) for these oligodendrocytes identified transcription factors mediating regulation in both discovery and replication data (replication hypergeometric $p = 9.08 \times 10^{-41}$). We identified *SOX8, SREBF1,* and *NKX6-2*

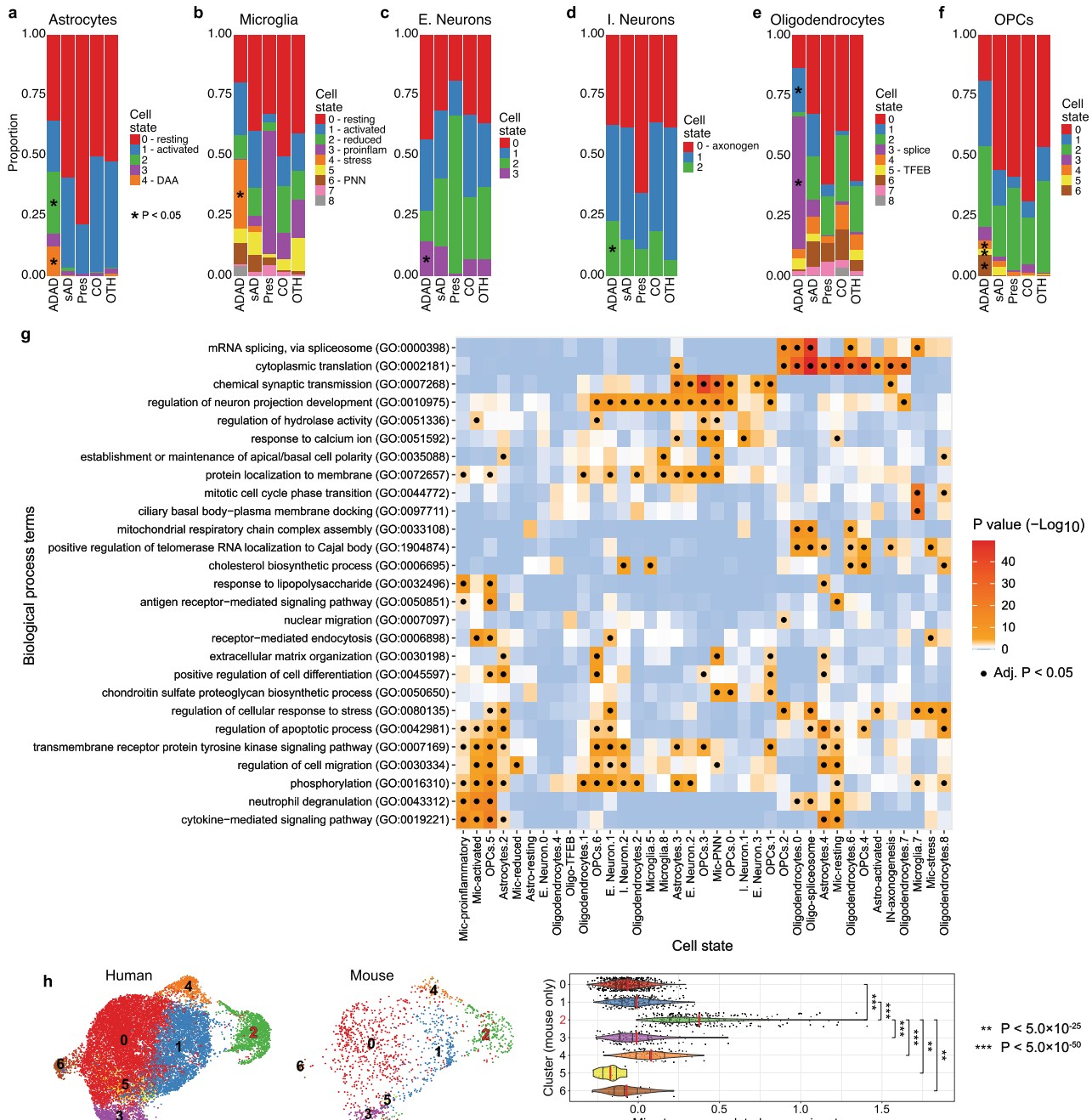

**Fig. 3 | ADAD participants exhibit distinct signatures in astrocytes, microglia, OPCs, oligodendrocytes, and neurons. a–f** Proportion plots show the enrichment of certain cell states in ADAD participants compared to all other participants. The proportion was calculated for each sample (see Methods). For visualization, sample proportions are averaged by AD status. (*) represents a significant ($p < 0.05$) enrichment of that cluster within ADAD samples as determined by linear regression. Exact $p$ values can be found in Supplementary Dataset 6. ADAD autosomal dominant AD, sAD sporadic AD, Pres presymptomatic, CO neuropath free, OTH non-AD neurodegenerative. **a** Astrocytes (Astro-DAA = cluster 4). **b** Microglia (Mic-stress = cluster 4). **c** Excitatory neurons. **d** Inhibitory neurons. **e** Oligodendrocytes (Oligo-spliceosome cluster 3). **f** OPCs. **g** A heatmap of the enriched pathways within

the upregulated genes for each cell state. The DEGs were isolated from the linear mixed models comparing each cell state to all other cell states of the same cell type. GO Biological Process terms were summarized and selected as described in the Methods. (·) indicates a significant (Benjamini–Hochberg $p < 0.05$) association as calculated by the R package enrichR. Exact $p$ values can be found in the Source Data file. **h** 5xFAD mouse validation of Mic-stress ADAD cluster (cluster 2 here). Left and middle: a UMAP of integrated microglia split by species. Right: a violin plot showing that mouse cells in the ADAD cluster have a higher human microglia ADAD cluster signature score than mouse cells in other clusters. (**) = $p < 5.0 \times 10^{-25}$, (***) = $p < 5.0 \times 10^{-50}$. Source data are provided as a Source Data file.

linked to myelination[17,52,53], *NFE2L2/NRF2* associated with multiple AD pathologies including Aβ, Tau, and oxidative stress[54], and *ZNF518A* associated with increased somatic-mutational burden in AD oligodendrocytes[55] (Fig. 4g and Supplementary Dataset 19). Downregulation of *SREBF1* was previously reported in oligodendrocytes in

AD brain tissue[17], and associated with regulation of *ErbB/mTOR* signaling pathways and autophagy[56,57]. Changes in autophagy markers were reported previously in microglia of p.R62H carriers[46] and our data suggest that autophagy dysregulation may also be a feature of oligodendrocytes in p.R62H carriers.

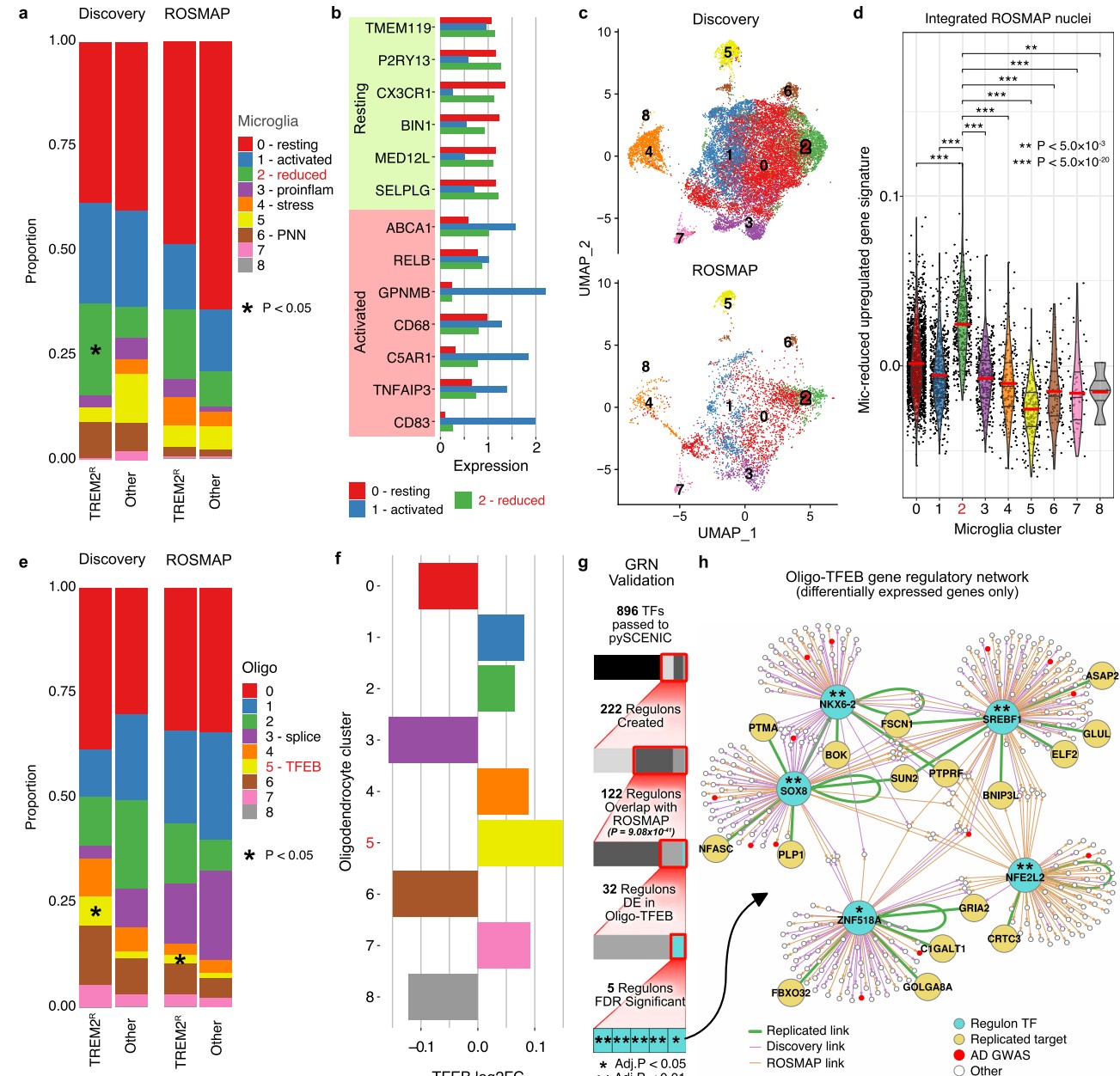

**Fig. 4 | *TREM2* reduced activation variant carriers (p.R47H, p.R62H, and p.H157Y) have distinct microglia and oligodendrocyte profiles. a–d** Microglia (Mic-reduced = cluster 2). **e–g** Oligodendrocytes (Oligo-TFEB = cluster 5). **a, e** Proportion plots show a significant ($p < 0.05$) enrichment (*) of cell states in *TREM2* reduced activation carriers (TREM2^R) compared to all other samples as determined by linear regression. Exact $p$ values can be found in Supplementary Dataset 6. The proportion was calculated for each sample (see Methods). For visualization, sample proportions are averaged by group. "Other" represents all sAD non-*TREM2* reduced activation carriers, including carriers of other *TREM2* variants. **b** Barplot shows the expression of both resting and activated microglia marker genes in Mic-reduced (mic.2) compared to the Mic-resting (mic.0) and Mic-activated cell states (mic.1). Expression was corrected for the age of death and sex using partial residuals. **c** UMAP plots showing the integrated nuclei from the discovery cohort and ROSMAP, split by cohort. **d** Violin plot of cell state expression signatures in the ROSMAP nuclei. The signature was calculated from the upregulated genes from the discovery cohort. Differences in signature scores were calculated using linear regression. (*) = $p < 0.05$, (**) = $p < 5.0 \times 10^{-3}$, (***) = $p < 5.0 \times 10^{-20}$; exact $p$ values can be found in the Source Data file. **f** Barplot shows the log2 fold-change of *TFEB* by oligodendrocyte cell state. **g** Identification of gene regulatory networks (GRN) in Oligo-TFEB discovery and replication cohorts. Regulons were filtered to include only those identified in both cohorts ($p = 9.98 \times 10^{-41}$; hypergeometric analysis) with significant differential expression in Oligo-TFEB. Then those regulons with significant (Benjamini–Hochberg $p < 0.05$; hypergeometric analysis and Benjamini–Hochberg multiple testing correction) coincidence in the underlying target genes between cohorts were selected. **h** Gene regulatory network for transcription factors (TF; shown in blue) replicated in both discovery (purple edges) and ROSMAP (orange edges) datasets for oligo-TFEB. Replicated target genes and edges are shown in yellow and green respectively. Genes within AD GWAS loci are highlighted in red. **g, h** (*) = Benjamini–Hochberg $p < 0.05$, (**) = Benjamini–Hochberg $p < 0.01$. Source data are provided as a Source Data file.

## MS4A resilience variant carriers show a specific inflammatory microglial activation state

Carriers of rs1582763-A, an intergenic allele associated with reduced risk for AD and higher CSF sTREM2 levels[14], showed increased proportions of nuclei in Mic.3 (Mic-proinflammatory) compared to non-carriers (generalized linear regression β = 0.15, $p = 1.67 \times 10^{-3}$; Fig. 5a; Supplementary Fig. 12 and Supplementary Datasets 6e, 9). The Mic-proinflammatory state displayed a proinflammatory profile, including the upregulation of

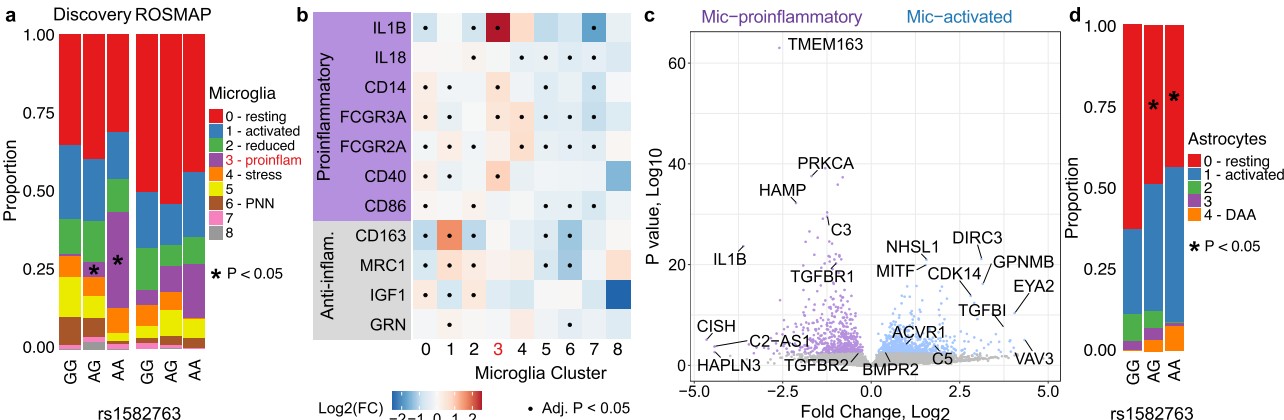

**Fig. 5 | Unique microglial and astrocytic signatures for *MS4A* rs1582763 carriers. a** Proportion plot shows a dose-dependent enrichment of Mic-proinflammatory (cluster 3) in carriers of the rs1582763-A allele. The proportion was calculated for each sample, and for visualization, sample proportions are averaged by the group. (*) represents a significant ($p < 0.05$) enrichment as determined by linear regression. Exact *p* values can be found in Supplementary Dataset 6. **b** Heatmap of proinflammatory (purple) and anti-inflammatory (gray) microglia marker gene log₂ fold changes (Log₂(FC)) within the microglia cell states. (·) indicates a significant (Benjamini−Hochberg $p < 0.05$) association as determined by linear regression and Benjamini−Hochberg multiple testing correction. Exact log fold-change and *p* values can be found in the Source Data file. **c** Volcano plot of DEGs determined by linear mixed models between the "main" (Mic-activated, blue) and "MS4A" (Mic-proinflammatory, purple) activated microglia clusters. **d** A proportion plot depicts a significant ($p < 0.05$) reduced proportion (*) of astro.0 (non-activated) in rs1582763-A carriers and a trend for the enrichment of astro.1 (activated) as calculated by linear regression. Exact *p* values can be found in Supplementary Dataset 6e. Source data are provided as a Source Data file.

*ILB1* (linear mixed effects regression log2FC = 3.45, Benjamini−Hochberg $p = 2.07 \times 10^{-36}$), *CD14* (log2FC = 0.63, Benjamini−Hochberg $p = 1.87 \times 10^{-3}$), *FCGR3A* (log2FC = 0.50, Benjamini−Hochberg $p = 3.31 \times 10^{-3}$), and *CD40* (log2FC = 0.91, Benjamini−Hochberg $p = 9.84 \times 10^{-3}$; Fig. 5b). Its upregulated genes were associated with "response to lipopolysaccharide" (Benjamini−Hochberg $p = 2.15 \times 10^{-6}$) and "cytokine-mediated signaling" (Benjamini−Hochberg $p = 9.65 \times 10^{-10}$; Fig. 3g). Mic-proinflammatory was present in the ROSMAP cohort (upregulation signature score of Mic-proinflammatory generalized linear regression *p* range $1.68 \times 10^{-235}$ to $9.98 \times 10^{-7}$; Supplementary Fig. 10), but it was not enriched within rs1582763 carriers ($p > 0.05$), possibly because the ROSMAP cohort contained few homozygous recessive carriers ($n = 3$) which were enriched in the discovery cohort by the design of the study ($n = 13$). In conjunction with increased proportions of Mic-proinflammatory, rs1582763-A carriers also exhibited a trend for decreased proportions of nuclei in Mic-activated (Supplementary Dataset 6e). Mic-activated had an upregulation of activated-microglia genes *CD68* (linear mixed effects regression log2FC = 0.39, Benjamini−Hochberg $p = 1.06 \times 10^{-4}$), *CD83* (log2FC = 1.34, Benjamini−Hochberg $p = 9.44 \times 10^{-37}$), *TNFAIP3* (log2FC = 0.57, Benjamini−Hochberg $p = 2.16 \times 10^{-7}$), *C5AR1* (log2FC = 1.23, Benjamini−Hochberg $p = 4.03 \times 10^{-46}$), *GPNMB* (log2FC = 1.73, Benjamini−Hochberg $p = 6.99 \times 10^{-102}$), and *ABCA1* (log2FC = 0.82, Benjamini−Hochberg $p = 1.78 \times 10^{-45}$; Fig. 4b, Supplementary Fig. 8), and a significant overlap with genes also upregulated in the DAM[38], MGnD[39], HAM[40], and aging[58] signatures (Supplementary Dataset 18).

We then identified the DEGs between Mic-activated and Mic-proinflammatory. Mic-activated showed increased expression of *C5* (linear mixed effects regression log₂FC = 1.79, Benjamini−Hochberg $p = 3.46 \times 10^{-3}$), whereas Mic-proinflammatory had increased expression of *C3* (linear mixed effects regression log₂FC = −1.24, Benjamini−Hochberg $p = 4.98 \times 10^{-26}$; Fig. 5c and Supplementary Dataset 20a). Previous analyses have indicated a protective role for *C3* in AD and a potentially detrimental role for *C5*[59]. This is consistent with the protective effect of rs1582763-A. Mic-activated also had increased *ACVR1* (linear mixed effects regression log₂FC = 0.97, Benjamini−Hochberg $p = 2.92 \times 10^{-3}$) and *BMPR2* (log₂FC = 0.39, Benjamini−Hochberg $p = 3.54 \times 10^{-2}$), indicating increased BMP signaling, whereas Mic-proinflammatory had high *TGFBR1* (linear mixed effects regression log₂FC = −0.98, Benjamini−Hochberg $p = 4.82 \times 10^{-17}$) and *TGFBR2* (log₂FC = −0.35, Benjamini−Hochberg $p = 4.50 \times$

$10^{-2}$), indicating increased TGF-β signaling (Fig. 5c and Supplementary Dataset 20a). These genes encode for related receptors in the TGF-β superfamily, which is implicated in multiple neurological disorders[60]. Mic-proinflammatory also showed increased expression of *TMEM163*, an AD GWAS gene[61]. *TMEM163* is involved in transporting zinc into cells where the zinc influences reactive oxygen species (ROS) levels[62]. Evidence suggests ROS causes genomic damage to neurons leading to cell death in AD[63]. A gene ontology analysis also showed an upregulation of genes related to "cytokine response/production" (Benjamini−Hochberg $p = 2.89 \times 10^{-9}$) and "regulation of ERK1 and ERK2 cascade" (Benjamini−Hochberg $p = 2.37 \times 10^{-7}$; Supplementary Dataset 20b). In addition, carriers of rs1582763-A had decreased proportions of Astro-resting (Astro.0) and a trend towards increased proportions of Astro-activated (Astro.1) (Fig. 5d, Supplementary Fig. 7f, and Supplementary Dataset 6e). The MS4A genes are not expressed in astrocytes, suggesting cellular crosstalk synchronizes microglia and astrocytes' activation.

## *APOEε4* carriers show vulnerability to ferroptosis in inhibitory neurons

We identified a microglial and an inhibitory neuron cell state with reduced proportions within *APOEε4* carriers compared to non-carriers. The microglia state, Mic.6 (Mic-PNNs; generalized linear regression β = −0.13, $p = 1.85 \times 10^{-2}$; Supplementary Dataset 6f, 9c), showed upregulation of genes relating to "extracellular matrix organization" (Benjamini−Hochberg $p = 6.68 \times 10^{-6}$, Fig. 3g) and "chondroitin sulfate proteoglycan biosynthetic process" (Benjamini−Hochberg $p = 5.03 \times 10^{-4}$, Fig. 3g) pointing to perineuronal nets (PNNs) and plasticity[64]. The neuronal state, IN.0 (IN-axonogenesis; generalized linear regression β = −0.09, $p = 8.06 \times 10^{-3}$), showed upregulation of genes involved in "cytoplasmic translation" (Benjamini−Hochberg $p = 5.91 \times 10^{-21}$, Fig. 3g) and "axonogenesis" (Benjamini−Hochberg $p = 2.00 \times 10^{-6}$, Supplementary Dataset 17). Within IN-axonogenesis, *APOEε4* carriers had upregulation of genes *PRNP* and *GPX4* related to "Ferroptosis" (Benjamini−Hochberg $p = 1.11 \times 10^{-2}$, Supplementary Dataset 21g) a pathology often found in AD samples and mouse models[65-67]. This is concordant with the recent finding in *APOE*, suggesting that the *APOEε4* protein variant has a reduced capacity to inhibit iron release by ferritin and prevent the accumulation of intracellular iron and lipid peroxides, which lead

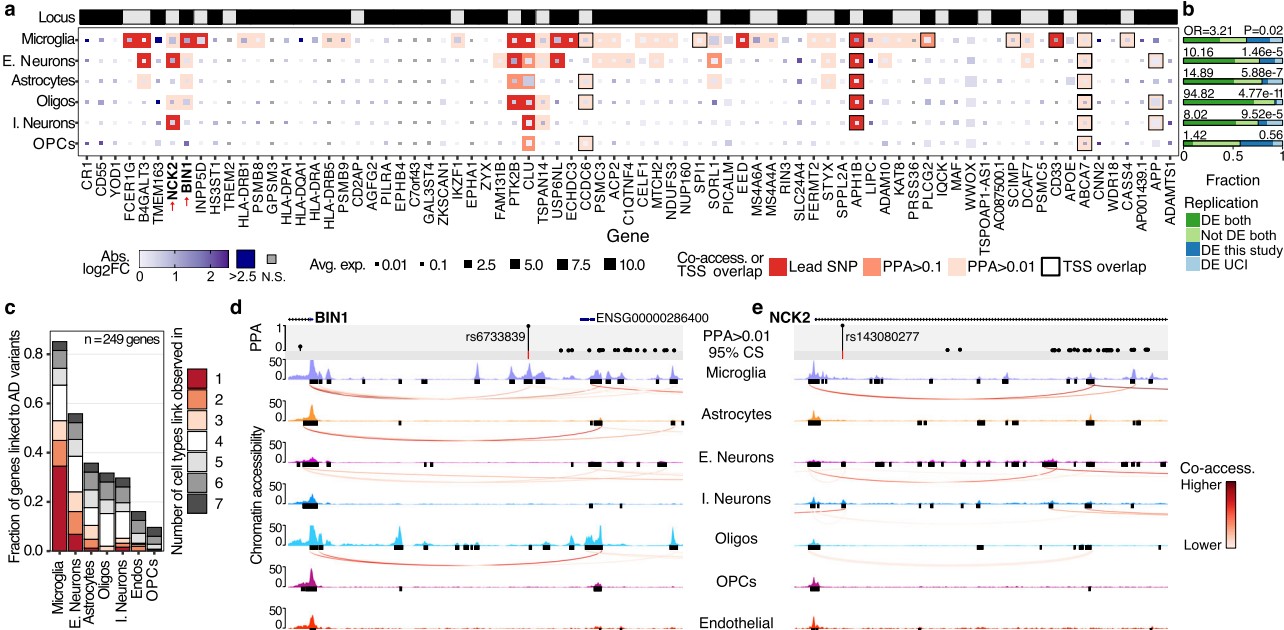

**Fig. 6 | Transcriptional states complement chromatin accessibility to help prioritize causal cell types and genes mediating AD GWAS risk variants.**
**a** Overview of prioritized risk genes by AD GWAS studies and their transcriptional changes across cellular states (absolute log$_2$ fold-change in any cellular state for cell type). The "Locus" row indicates genes within the same locus using alternating black and gray rectangles. The color of the squares represents the max log$_2$ fold-change of the gene between cell states (subclusters) of that cell type (gray: not significant). The square size represents the average log10 transformed gene expression. Borders represent co-accessibility (red background) or overlap (black outline) between the TSS and a regulatory element containing a prioritized (95% credible set) AD variant. Background color intensity corresponds to the highest posterior probability of association (PPA) of the 95% credible set variants

overlapping the TSS or co-accessible element. **b** Replication of the parietal lobe differential expression results using the UCI prefrontal cortex snRNA-seq data. OR: Fisher exact test odds-ratio (replicated vs. non-replicated). **c** Distribution of genes co-accessible or with their TSS overlapping a regulatory element (snATAC-seq narrow peak) containing a fine-mapped AD risk genetic variant. The color indicates the number of cell types the co-accessibility signal was detected in. **d** Chromatin accessibility signals across cell types for the BIN1 locus. The lead variant is represented by a red vertical bar, and the fine-mapping PPAs are plotted for each variant with PPA >0.01. TSS regions co-accessible with variant-overlapping regulatory elements are plotted as arcs below each signal track. **e** Same visualization as (d) for the NCK2 locus. Source data are provided as a Source Data file.

---

to ferroptosis[68]. Therefore, ferroptosis could explain the reduction of *APOEε*4+ nuclei in this cell state.

*APOEε*4 carriers also showed an upregulation of genes involved in 'ribosome biogenesis' (Benjamini–Hochberg $p = 1.42 \times 10^{-2}$) and "mitotic G1/S transition checkpoint signaling" (Benjamini–Hochberg $p = 1.42 \times 10^{-2}$) in microglia, "maintenance of protein localization in ER" (Benjamini–Hochberg $p = 7.75 \times 10^{-3}$) and "COPI coating of Golgi vesicles" (Benjamini–Hochberg $p = 7.75 \times 10^{-3}$) in astrocytes, "regulation of nuclear division" (Benjamini–Hochberg $p = 3.12 \times 10^{-2}$) in oligodendrocytes, and "cytoplasmic translation" in OPCs (Benjamini–Hochberg $p = 4.34 \times 10^{-5}$; Supplementary Dataset 21).

### Genes within risk AD GWAS loci show variable expression between transcriptional states

GWAS meta-analyses have successfully identified genomic loci associated with sAD[11–13]. We sought to leverage the glial and neuronal cell states to help map genomic loci implicated in the general AD population onto genes and cell types. This is based on the rationale that genes differentially expressed between cell states might be functionally relevant to that cell type regardless of the relative expression of the gene compared to other cell types. We curated a list of genes in loci identified in recent AD GWAS, starting with 89 genes previously prioritized[13,61,69] (77 measured in this study) from 38 different loci (Supplementary Dataset 22a); we extended the list by adding the "non-prioritized" genes within these AD GWAS associated loci also present in this dataset (530 genes; Supplementary Dataset 22b, c).

Out of the 77 prioritized genes, 68 (from 37 loci) showed differential expression between the cell states of at least one cell type (Fig. 6a and Supplementary Fig. 13). We also found that for 13 genes,

the cell type with the greatest expression level variability (log fold-change between cell states) did not match the cell type with the highest average expression. For example, *CR1* showed the highest average expression in oligodendrocytes, but microglia and OPCs had larger log$_2$ fold changes in expression (log$_2$FC) between cell states. This suggests that transcriptional states can provide an additional layer of information to help map GWAS genes to cell types. Rare coding variants in *PLCG2*, implicated in *TREM2*-dependent microglial function in AD[70], have been recently reported to protect against AD[71]. We found differential expression of *PLCG2* between cell states within all cell types, suggesting an extended functionality beyond the *TREM2* signaling pathway in microglia (Fig. 6)[72,73]. These PLCG2 expression differences were largely driven by the significant overexpression detected in ADAD brains across all cell types except microglia, which was nominally significant (Supplementary Dataset 10).

We accessed snRNA-/snATAC-seq data from 20 additional brains (prefrontal cortex) from the UCI MIND's ADRC[17]. Using the snRNA-seq data, we replicated the expression and fold-change patterns identified in the discovery cohort (see Methods). Specifically, we observed a high correlation between cell-type-specific gene expression (Pearson $R = 0.93$; $p = 5.65 \times 10^{-211}$; Fig. 6b) and an overall concordance in the DEG by cell-type calling (Fisher exact test OR = 7.88, $p = 4.09 \times 10^{-24}$; Supplementary Fig. 14 and Supplementary Dataset 23) for the prioritized genes between cohorts. All cell types, except OPCs, individually had a significant concordance in DEG calling.

Next, we sought to determine if genetic variants identified in AD GWAS showed chromatin co-accessibility with the prioritized genes, thus providing an independent layer of evidence relating genetic variants to genes and cell types. We employed the snATAC-seq from UCI[17]

and fine-mapped AD GWAS credible sets[61] to analyze cell-type-specific chromatin accessibility between the transcription start sites (TSS) of the prioritized genes and the variants in the AD GWAS credible sets. We determined that microglia cells had the highest fraction of genes co-accessible with AD risk variants, as was previously reported[17]. A large fraction (34.5%) of these genes only had co-accessibility signals within microglia, indicating that they likely mediate AD genetic risk in a cell-type-specific manner (Fig. 6c). For example, rs6733839 in the *BIN1* locus is co-accessible with the *BIN1* TSS only in microglia (Fig. 6d). This is concordant with our predictions using transcriptional data (Fig. 6a and Supplementary Dataset 8), which showed that *BIN1* is under-expressed in Mic-activated compared to both Mic-resting and Mic-proinflammatory (*MS4A* associated), suggesting that the role of *BIN1* in AD risk is dependent on microglia activation.

A recent study suggested that microglial cells mediate AD genetic risk at the *NCK2* locus (lead variant rs143080277, MAF = 0.0054), principally based on the higher expression of *NCK2* in microglia[74]. We also observed that *NCK2* expression is highest in microglia (Fig. 6a and Supplementary Fig. 12); however, we identified the largest *NCK2* expression differences within excitatory neuron cell states (max. linear mixed effects regression fold-change = 2.02). Analyses of chromatin accessibility show that rs143080277 is co-accessible with the *NCK2* promoter in excitatory and inhibitory neurons, but not microglia (Fig. 6e). Altogether, these results highlight that DE between cell-type-specific transcriptional states provides additional information to help prioritize target genes and cell types mediating genetic risk at non-coding loci, which is complementary to other evidence, including cell-type-specific chromatin accessibility.

## Discussion

We performed snRNA-seq on nuclei from the understudied parietal cortex from a cohort enriched in carriers of AD genetic variants in *APP*, *PSEN1*, *TREM2*, *APOE*, and the *MS4A* cluster (rs1582763-A, an intergenic resilience variant in the *MS4A* locus associated with increased levels of soluble *TREM2* in CSF and reduced AD risk[14]).

In this cross-sectional study, we identify genes commonly dysregulated across AD groups, including *FARP1*, *SQSTM1*, *CNTNAP2*, *LPL*, and *SNTG* in microglia, astrocytes, OPCs, oligodendrocytes and excitatory neurons respectively. These represent core genes and pathways perturbed across a wide range of genetic backgrounds and are, therefore, powerful potential targets for therapeutic intervention. We also noted that the parietal lobes of sporadic and autosomal dominant AD brains show similar transcriptional dysregulation in general, but the effect is usually stronger in ADAD. This supports the premise of the amyloid hypothesis, that the molecular changes in ADAD are also found in sAD, and shows that the ADAD cohorts provide unique opportunities to study the underlying biology of the disease.

It is worth noting that we observed cell states enriched within ADAD samples for all cell types, with the Mic-stress state almost exclusively present in these carriers. We cannot rule out the possibility that this specificity could be due to younger ages in ADAD (an average of 32.9 years younger) compared to sAD. However, upon integrating microglia and oligodendrocyte snRNA-seq data from the parietal cortex with that from the DLPFC, we observed that this transcriptional state is also present in sporadic late-onset AD (Fig. 4c, g). In sAD, the parietal cortex is affected in the later stages of disease progression, whereas the DLPFC is affected earlier[75,76]. Therefore, the cell states identified in the parietal cortex of ADAD participants could represent accelerated stages of pathology like the DLPFC in late-onset sAD, consistent with the elevated tau PET found in the parietal region of ADAD compared to sAD patients[77]. Additional studies capturing multiple brain regions with varying degrees of pathology will provide the data to understand these observations in more detail.

*TREM2* p.R47H, p.R62H, and p.H157Y show reduced cell activation in vitro[20,45]. Here, we report microglia and oligodendrocyte cell

states associated with *TREM2* risk variant carriers. *TREM2* is mainly expressed in microglia. However, loss of function mutations in *TREM2* cause Nasu-Hakola disease, which is characterized by white matter changes, including loss of myelination, suggesting oligodendrocyte-microglia crosstalk[78]. Therefore, it is likely that AD patients with *TREM2* variants had altered microglia behavior which changed the microenvironment, driving the oligodendrocyte cell state found in *TREM2* variant carriers. The microglial cluster associated with these reduced activation variant carriers displayed a resting-like state, suggesting they might benefit from treatments targeting *TREM2* to prevent its cleavage, increasing microglial activation[79]; however, not all *TREM2* risk variants are functionally equivalent[14,80,81]. We also observed a microglia state expressing activation markers enriched for homozygous carriers of the *MS4A* resilient variant. In contrast to the major activated microglia state, this cell state showed upregulation of proinflammatory genes and cytokine signaling. A better understanding of this microglia resilience state could improve efforts to induce therapeutic microglial activation.

*APOE*ε4 carriers had decreased proportions of nuclei in Mic-PNN and IN-axonogenesis, expression states that both influence axon regeneration and brain plasticity. The disruption of these cell states could be a contributor to the faster cognitive decline associated with the ε4 allele. Pathway analysis implicates ferroptosis in the loss of IN-axonogenesis cells suggesting this pathway warrants further investigation in *APOE*ε4 carriers.

Most AD GWAS hits are non-coding variants that potentially influence gene expression. By inspecting the cell state differential expression results, we found that multiple cell types might mediate some genetic loci's effect. For instance, both differential expression and co-accessibility analyses linked the *NCK2* GWAS hit to neurons in addition to the initially proposed microglia. This example highlights the benefit of using cell state DE as an additional layer in linking GWAS signals to genes and cell types.

These analyses are limited by the rarity of mutations in *APP*, *PSEN1*, and the low frequency of variants in *TREM2* in the general population. As a result, samples with mutations in *APP* and *PSEN1* were merged and considered as a single ADAD group and three *TREM2* variants (p.R47H, p.R62H, and p.H157Y) were merged as the *TREM2*-reduced activation group despite there being slight differences in the functional mechanisms. Further analyses on additional tissue samples from carriers of these variants are needed to fully uncover the variant-specific effects in these critical AD genes.

Disease heterogeneity could explain the failure of many promising clinical trials to meet their endpoints[82]. The mechanisms driving AD heterogeneity are complex and rarely studied. Here, we show that genetic variants influence cell expression states, and, therefore, could explain some disease heterogeneity. The molecular characterization of the genes and pathways driving these cell states elucidates the functional mechanisms driving disease heterogeneity and possible targets for therapeutic intervention in the era of personalized medicine.

In conclusion, our findings support that single AD risk variants can influence the transcriptional landscapes of multiple brain cell types. Pathogenic mutations in *APP* and *PSEN1* altered the profiles of neurons, but more especially glia when compared to controls and sAD. *TREM2* risk variants shifted microglial and oligodendrocytic profiles and the *MS4A* resilience variant inflated a proinflammatory microglia profile. Each of these changes can modify AD's pathological progression and clinical manifestations.

## Methods

### Processing of brain tissue samples
The Washington University Translational Human Neurodegenerative Research (THuNDR) laboratory, which serves as the Neuropathology Core for the Knight Alzheimer Disease Research Center (Knight ADRC)

and the Dominantly Inherited Alzheimer Network (DIAN) study, provided the postmortem parietal lobe tissue samples. These samples were obtained with informed consent for research use and approved by the Washington University School of Medicine in St. Louis Institutional Review Board. According to the National Institute on Aging-Alzheimer's Association (NIA-AA) criteria, AD neuropathological changes were assessed. Their demographic, clinical severity, and neuropathological information are presented in Table 1. RIN scores were used to evaluate bulk RNA quality for each sample before snRNA-seq library preparation[83] (Supplementary Dataset 24).

### Nuclei isolation and snRNA-seq on the 10X Genomics platform

The frozen human parietal cortex samples were processed according to the "Nuclei extraction and library preparation" protocol described by ref. 23. Briefly, the tissue was homogenized, and the nuclei were isolated using a density gradient. The nuclei were then sequenced using the 10X Chromium single cell Reagent Kit v3, targeting 10,000 cells per sample and 50,000 reads per cell for each sample.

### SnRNA-seq data processing with 10X genomics CellRanger software and data filtering

The CellRanger (v3.0.2 10X Genomics) software was employed to align the sequences and quantify gene expression. We used the GRCH38 (release-93) reference to prepare a pre-mRNA reference according to the steps detailed by 10X Genomics (References- 3.0.0). The software was packaged into a Docker container (https://hub.docker.com/r/ngicenter/cellranger3.0.2), allowing us to launch it within the McDonnell Genome Institute (MGI) infrastructure, reducing the computing time for generating the BAM files. Four samples failed in library prep and sequencing, leaving 70 samples to pass to QC.

Filtering and QC were done using the Seurat package (v3.1.2) on each subject individually. Each raw gene expression matrix for each sample was plotted using *BarcodeInflectionsPlot* to calculate the inflection points derived from the barcode-rank distribution. Thresholds were selected to isolate uniform regions of the distribution (Barcode-rank Distribution, Table 2, Supplementary Fig. 15, and Supplementary Dataset 25). Once the thresholds were determined, a subset of the data were isolated. We removed nuclei with high mitochondria gene expression following the dynamic model proposed by ref. 19. Briefly, the nuclei were grouped by their percentage of mitochondria values using k = 2 clustering, and the group with the higher percentage values was removed. Genes not expressed in at least ten nuclei were removed from the final matrix. To detect and discard doublets, we used DoubletFinder[84] (v1.0.0), which removes nuclei with expression profiles that resemble synthetically mixed nuclei from the dataset. The gene expression matrices from all samples were combined in R independently for further processing using the Seurat protocol. One sample was removed during this process due to low nuclei counts, leaving 69 samples.

### Dimensionality reduction, clustering, and UMAP

The merged expression matrix was normalized using the *SCTransform* protocol by Seurat. This function calculates a model of technical noise in scRNA-seq data using "regularized negative binomial regression" as described previously in ref. 85. We regressed out, during the normalization, the number of genes, the number of UMIs, and the percentage of mitochondria. The principal components were calculated using the first 3000 variable genes, and the Uniform Manifold Approximation and Projection (UMAP) analysis was performed with the top 14 PCs. The clustering was done using a resolution of 0.2.

### Cluster annotation and quantification of regional and individual contributions to cell types

We employed a list of marker genes we had previously curated[23] to annotate brain snRNAseq data. We used the *DotPlot* function (Seurat

package) to visualize the average expression of genes related to specific cell types. This approach enabled the labeling of cell types based on the overall expression profile of the nuclei, regardless of dropout events. In addition, we employ a supervised method termed Garnett[86] (v0.1.14) that leverages machine learning to classify each nucleus and estimate cluster homogeneity. This method also provides a metric of gene ambiguity, which enables further optimization of the marker genes to be included in the classification process. For this method, we employed *SYT1, SNAP25, and GRIN1* to classify neurons, *NRGN, SLC17A7*, and *CAMK2A* for excitatory neurons and *GAD1* and *GAD2* for inhibitory neurons; *AQP4* and *GFAP* for astrocytes; *CSF1R, CD74*, and *C3* for microglia; *MOBP, PLP1*, and *MBP* for oligodendrocytes; *PDGFRA, CSPG4*, and *VCAN* for oligodendrocyte precursor cells (OPCs); *CLDN5, TM4SF1*, and *CDH5* for endothelial cells and ANPEP for pericytes. We employed the function *check_markers* (Garnett package) to evaluate the ambiguity score and the relative number of cells for each cell type. A classifier was then trained using the marker file, with "num_unknown" set to 50. This classifier annotates cells with cell-type assignments extended to nearby cells using the "clustering-extended type" labeling option. At this stage, one ambiguous cluster and one subject-specific cluster were dropped. One sample was predominantly in the ambiguous cluster, and another was predominantly in the subject-specific cluster, so those samples were removed entirely, leaving 67 samples. Distributions of UMI counts, gene counts, and percentage of mitochondrial reads for each sample are shown in Supplementary Fig. 16.

### Identification of alternative cell-type transcription states

The nuclei within the primary cell-type clusters were each isolated from the full dataset and re-clustered. We re-normalized the data subset using the same protocol as explained in section 4 in methods. The number of PCs used for UMAP dimensionality reduction was different for each cell type, 4, 8, 10, 6, and 5, for neurons, oligodendrocytes, microglia, astrocytes, and OPCs, respectively. We then employed Seurat's *FindNeighbors* and *FindClusters* functions to identify unique cell states or subclusters (resolution = 0.1, 0.2, 0.2, 0.05, and 0.15). Additionally, we used the Garnett protocol to examine nuclei in each expression state within each cell type to detect and remove those nuclei that did not resemble a trustworthy expression profile from downstream analyses. After this final stage of QC, we ended with 67 of the 74 brains and 294,114 of the 1,102,459 nuclei. Distributions of UMI counts, gene counts, and percentage of mitochondrial reads by cell state are shown in Supplementary Fig. 17.

### Sample and batch entropy by clusters and cell state

To evaluate the sample and batch effects in clustering, the sample and batch entropies were calculated for clusters and cell types for the full snRNA-seq dataset and the cell states for each cell type[87]. Shannon entropy was used to calculate individual cluster entropies (Eq. 1), and the weighted sum was used to calculate overall clustering entropy (Eq. 2)[88]. The entropies were normalized by dividing each entropy value by the maximum entropy possible for each scenario (batch (n = 14) = 3.81; sample (n = 67) = 6.07) (Supplementary Dataset 5).

$$H(i) = -\sum_{j \in K} p\left(i_j\right)\log_2 p(i_j) \tag{1}$$

$$H = \sum_{i \in C} H(i)\frac{N_i}{N} \tag{2}$$

### Excitatory and inhibitory neuron classification

Neuronal marker genes *SLC17A7* and *GAD1* from ref. 25. were used to classify the individual neuron cell states as either excitatory (EN) or inhibitory (IN), respectively (Supplementary Fig. 18). Expression of

these genes was log2 transformed, averaged by sample within each cluster, and then averaged by cluster to get a final score for each cluster. Downstream analyses were performed within these new classifications. Gene markers for the cortical layer were also from Lake et al. to link each neuronal cell state to a probable cortical layer (Supplementary Fig. 18).

## Differential proportion analysis

We employed linear regression models testing each individual's cell state compositions to identify associations between cell-type transcriptional states and disease groups (ADAD, sAD, *TREM2*, *TREM2*_reduced, rs158276). More explicitly, the number of nuclei a subject had in a specific cell

$$\text{Proportion} = \frac{\text{numberNuclei(state}_i)}{\text{totalNuclei(cellType)}} \quad (3)$$

state was divided by the subject's total nuclei count for that cell type creating a proportion (Eq. 3). The proportions were normalized using a cube root transformation and were corrected by sex, age of death, and disease status depending on the variable of interest. We removed participants who contributed fewer than 60 nuclei to the cell type cluster. The *TREM2* and *APOE* analyses only included the 31 sAD samples. We utilized *glm*, a standard function in R, to implement the model (Eqs. 4–8).

$$\text{rs1582763(Additive)}: \quad (\text{Proportion})^{1/3} \sim \text{Genotype} + \text{Sex} + \text{ADstatus} \quad (4)$$

$$\textit{APOE}\varepsilon4 +: \quad (\text{Proportion})^{1/3} \sim \text{APOEstatus} + \text{Sex} + \text{AOD} \quad (5)$$

$$\begin{aligned}\textit{TREM}2, \\ \textit{TREM}2-\text{reduced}:\end{aligned} \quad (\text{Proportion})^{1/3} \sim \text{TREM2status} + \text{Sex} + \text{AOD} \quad (6)$$

$$\text{ADAD}: \quad (\text{Proportion})^{1/3} \sim \text{Sex} + \text{ADstatus} \quad (7)$$

$$\text{sAD}: \quad (\text{Proportion})^{1/3} \sim \text{Sex} + \text{ADstatus} \quad (8)$$

To visualize these results, cell state proportions were averaged between the samples in a group and displayed in a stacked barplot using the ggplot2 (v3.3.6) library in R. The location and density of nuclei for these groups in the UMAP space were also visualized using a modified version of SCANPY's *embedding_density* functions that extended its functionality to quantitative variables (rs1582763 allele counts). This modified code can be found at https://github.com/HarariLab/parietal-snRNAseq[89].

## Cell state differential expression

To determine if there was unique functionality or potentially altered cell states due to disease, we fitted a linear mixed model that predicted the expression level of each gene for the individual nuclei by cell state and corrected for the subject of origin and sex (Eq. 9)[90]. Control, sAD, and ADAD samples were used to calculate cell-state differentially expressed genes. Expression levels were extracted from the Seurat objects using *GetAssayData* with the 'slot' parameter set to "counts". Age of death (AOD) was not included in the model because AOD is correlated with ADAD status. The R package nebula[91] (v1.1.5) was used to implement the model, including parameters for a zero-inflated negative binomial distribution (model = "NBLMM", method = "LN") and the random effect of the subject of origin[91]. The number of UMI's per nuclei was already accounted for during *SCTransformation*, so the

model did not need to account for the number of UMI's.

$$\begin{aligned}\text{Expression} \sim \text{CellState} + \text{Sex} + (1|\text{Subject}); \\ \text{ZI model} = \sim 1; \text{ZINB distribution}\end{aligned} \quad (9)$$

Within each cell type, differential expression (DE) was calculated on each cell state versus all other states and each state against each other state individually. Participants with sufficient nuclei counts (astrocytes, microglia: $n > 50$; excitatory neurons, inhibitory neurons, oligodendrocytes: $n > 60$; OPCs: $n > 66$) in the cell type cluster were included in the analysis. Thresholds were determined by natural breaks in the count distributions for each cell type.

## Genetic factor differential expression

Differentially expressed genes were identified within each cell type by genetic status, namely ADAD, TREM2, or sAD vs. Controls and *APOE*ε4− vs. *APOE*ε4+. The nuclei were isolated for each group (ADAD, TREM2 variant carriers, sAD, and *APOE*ε4+). Each group was compared to controls (neuropath. free controls or *APOE*ε4−) using linear mixed models as explained above. Samples with sufficient nuclei counts (astrocytes, microglia: $n > 50$; excitatory neurons, inhibitory neurons, oligodendrocytes: $n > 60$; OPCs: $n > 66$) in the cell type cluster were included in the analysis. The following model was used:

$$\begin{aligned}\text{Expression} \sim \text{GeneticStatus} + \text{AOD} + \text{Sex} + (1|\text{Subject}); \\ \text{ZI model} = \sim 1; \text{ZINB distribution}\end{aligned} \quad (10)$$

Age of death (AOD) was not included when analyzing ADAD. This model was run on the entire cell type and each cell state within the cell types. Only non-TREM2 sAD samples were included in the *APOE*ε4+ analysis. Only carriers of *TREM2* variants p.R47H, p.R62H, and p.H157Y were included in the *TREM2* compared to controls analyses.

## Handling of genetically related individuals

The data generated is primarily from unrelated donors, but also includes data from related samples from three nuclear families. In more detail, two pairs from two sibships from the Knight ADRC, and one pair of related donors from the DIAN cohort (sAD-family, sAD-presymptomatic family, and ADAD-family; Supplementary Dataset 26). To confirm that this would not bias the differential expression results, we ran analyses with both samples in a family retained and one sample from the family removed. Using the largest (oligodendrocytes) and one of the smallest (microglia) cell types, we tested ADAD compared to controls (ADAD-family, sample40 dropped) and sAD compared to controls (sAD-family, sample59 dropped) and calculated the correlation of the estimates and −log10-transformed $p$ values between the "retained" and "removed" analyses using *cor.test* in R (Supplementary Fig. 19).

## Overlapping genes

Overlapping DEGs from the genetic factor differential expression analyses were identified using two approaches. First, the results were grouped by genetic factor, and the cell types were overlapped (Supplementary Fig. 5 and Supplementary Dataset 13). Second, the results were grouped by cell type, and the genetic factors were overlapped (Supplementary Fig. 5 and Supplementary Dataset 15). Overlapping gene sets that included three or more overlaps were run through gene enrichment analysis using enrichR (v2.1; hosted by the Ma'ayan Laboratory[92,93]) (Supplementary Datasets 14, 16). Overlaps were visualized using the ComplexUpset (v1.3.3) library in R.

## Strength and direction of effects

To see which genes are simply more strongly differentially expressed in ADAD samples versus those unique to ADAD samples, the multiple-test-corrected significant DEGs for sAD, TREM2, and ADAD samples

compared to controls were isolated. The three gene lists were merged (union), and the estimates for each gene from each comparison were extracted. Distributions of the estimates split by sAD, *TREM2*, and ADAD were plotted for each cell type's nominally significant genes in each group. Heatmaps were created using the top 500 strongest estimates for each cell type (Supplementary Fig. 3). The heatmaps were visually inspected for gene modules, and the modules were run through gene enrichment analysis using enrichR (Supplementary Datasets 12, 27). These heatmaps were summarized to 25 genes in Fig. 2 by taking the genes with the strongest estimates from each module while maintaining the correct proportion of genes for each module.

The effects between sAD vs. controls and ADAD vs. controls were also directly compared by cell type using the results from the 'Genetic factor differential expression' analyses described above. Only the analyses run on the entire cell type population were interrogated. The results of these analyses were filtered to include only genes with nominal associations ($P < 0.05$) in both analyses to ensure that there could be confidence in the effect sizes for accurate comparison. The correlation coefficient was calculated using R's native *cor* function.

## Pathway analyses

The upregulated genes identified for each cell state were used in a subsequent pathway analysis. We used the R-based application enrichR. We used gene sets to determine pathway enrichment using the "KEGG 2021 Human" or "GO Biological Process 2021" gene sets. Downregulated genes were also run in the *APOE*-high neuron analysis.

A heatmap was created to summarize the upregulated "GO Biological Process 2021" (GO_BP) hits for all cell states. All GO_BP term -log10 *P* values for each cell state were merged into a single table. The union of the ten highest GO_BP terms for each cell state were then ranked (Rank) by finding the averaged transformed *P* value across all cell states for each term. The terms were then loaded into rrvgo (v1.6.0), an R package that implements the Revigo[94] tool for summarizing GO_BP terms. The function *calculateSimMatrix* was used to calculate the relationships between the GO_BP terms with variable inputs: orgdb = "org.Hs.eg.db", ont = "BP", method = "Rel". The terms were then summarized using reduceSimMatrix and the following variables: score = "Rank", threshold = .6, orgdb = "org.Hs.eg.db". The summarized terms or "parentTerms" and their *P* values were then used to make the heatmap. Rows were ordered using dist function method = "euclidean", and columns were ordered using method = "*p*" for Pearson correlation. Additional GO_BP terms were manually removed that showed similar signatures across the cell states and implied the same biological processes (compare Fig. 3g with Supplementary Fig. 6).

## Microglia expression states

We collected 12 gene sets (Aging[58], Homeostatic[95], Lipid-droplet-accumulating[96], Neurodegenerative[39], Proliferative-region-associated[97], Injury-responsive[98], Activated-response[99], Interferon-response[99], Human-Alzheimer's[40], Disease-associated[38], *TREM2*[79], and Granulin[79] microglia) associated with different microglia functional states that had been described in the literature. Each set was split into its up and downregulated gene lists. A hypergeometric test was performed using R's native *phyper* function to identify which previously reported microglial transcriptional states were recapitulated in this dataset.

## Neuronal *APOE* and *MHC-I* coexpression

We followed the methods outlined by ref. 100. Briefly, MAGIC[101] (v3.0.0) was used to impute gene expression, *APOE* expression greater than two standard deviations marked *APOE*-high expression, and the genes *HLA-A, HLA-B, HLA-C, HLA-E, HLA-F*, and *B2M* were summed to represent *MHC-I* expression. A total of 61 samples were analyzed (controls = 9, presymptomatic = 2, sAD = 28, ADAD = 15, others = 7). The correlation between *APOE* and *MCH-I* expression was calculated

using the native *cor.test* function in R. Genes differentially expressed between *APOE*-high and *APOE*-low neurons were identified using linear mixed models as previously described (*nebula* R package; model: expression ~ *APOE*High + Sex). These DE analyses were performed only on the IN.0 and IN.2 cell states split by AD condition. Enriched pathways in the significantly upregulated genes were identified using enrichR as previously described.

## ADAD-specific microglia cluster validation

To confirm the biological existence of Mic-stress (Mic.4), a microglia expression state observed in ADAD samples, single microglia cells ($n = 1551$) from 5xFAD mice were collected as described by ref. 22. Protein-encoding mouse genes were converted to their human orthologs using biomaRt (v2.42.1). The data were normalized using *SCTransform*, regressing out numbers of genes and UMIs. Seurat's best practices workflow was followed to integrate the mouse and human microglia using the human microglia as the reference dataset. Seven clusters were assigned using *FindNeighbors* and *FindClusters* with the first eight principal components and a resolution of 0.2 as input. A total of 297 mouse cells and 1413 of the 1429 human Mic.4 nuclei were recaptured in the post-integration cluster 2. The mouse cells were then re-isolated. A human Mic.4 signature score was calculated for each mouse cell by running the 412 upregulated genes that passed multiple-test corrections into Seurat's *AddModuleScore*. The significant pairwise differences in cluster scores were calculated using the linear mixed model:

$$\text{moduleScore} \sim \text{cluster}, \text{ziformula} = \sim 0, \text{family} = \text{'gaussian'} \quad (11)$$

executed using *glmmTMB* (v1.0.1).

## TREM2-enriched and rs1582763-enriched microglial cluster validation

Human snRNA-seq data from ROSMAP samples (DLPFC)[20] were used to confirm the Mic-reduced (Mic.2) and Mic-proinflammatory (Mic.3) cell states, enriched for *TREM2* p.R47H, p.R62H, and p.H157Y and the rs1582763-A allele respectively. The ROSMAP data has 11 sAD, 11 *TREM2* R62H, and 10 control participants with 3986 microglia. The *TREM2* p.R47H samples produced by the contributors of the ROSMAP snRNA-seq data were analyzed using nanostring and, therefore, not used in replication. The microglia were isolated and normalized using Seurat's *SCTransform* function with "return.only.var.genes" set to FALSE and regressing out "nCount_RNA" and "nFeature_RNA". The microglia were then integrated using 3000 features in *SelectIntegrationFeatures*, *PrepSCTintegration*, our data as a reference in *FindIntegrationAnchors*, and *IntegrateData*. To identify which nuclei fell into our original clusters, the integrated data were clustered using the first ten principal components as input for *FindNeighbors* and a resolution of 15 in *FindClusters*. This shattered the data finding 147 clusters. We assigned each of these 147 clusters an 'original' identity by isolating our cohort of nuclei from the individual clusters and identifying the most common original ID. This ID was transferred to the ROSMAP nuclei, similar to a k-nearest neighbor classifier[102]. These cluster identities were mapped to the pre-integrated normalized ROSMAP data.

To measure the accuracy of our label transfer, cell state signature scores were calculated for each ROSMAP nucleus by running the significantly upregulated genes with estimates greater than 0.25 from each discovery cell state into Seurat's AddModuleScore (Supplementary Dataset 28). This same process was run for the downregulated genes with estimates less than −0.25. The significant pairwise differences between clusters were calculated using the linear mixed model:

$$\text{moduleScore} \sim \text{cluster} + (1|\text{subject}), \text{family} = \text{'gaussian'} \quad (12)$$

It was executed using the *lmer* function from the lme4 (v1.1-23) R package (Supplementary Fig. 10). The previously described 'Differential proportion analysis' methods were then followed to verify the enrichment of *TREM2* p.R62H nuclei in the ROSMAP Mic-reduced (Mic.2) cluster and rs1582763 nuclei in the ROSMAP Mic-proinflammatory (Mic.3) cluster.

## TREM2-enriched oligodendrocyte cluster validation

The two oligodendrocyte clusters (29,478 nuclei) in the ROSMAP data were integrated with our oligodendrocytes and labeled with our original cluster identities as described above. A total of 264 clusters were identified using eight PCA dimensions at a resolution of 15. As described above, the upregulated and downregulated signature scores were calculated on the ROSMAP nuclei using the cell state DEGs identified from the discovery cohort. Pairwise significant differences in the signature score and enrichment of *TREM2* p.R62H variant carriers were calculated as described above (Supplementary Fig. 11). The previously described 'Differential proportion analysis' methods were followed to verify the enrichment of *TREM2* p.R62H nuclei in the ROSMAP Oligo-TFEB (Oligo.5) cluster.

## Reconstruction of gene regulatory network from snRNA-seq

We employed the python implementation of the SCENIC[103] analysis method called pySCENIC[104] (version 0.12.1) to evaluate the gene regulatory networks (GRN). We selected 8561 genes expressed in ≥5% of the full oligodendrocyte population in the discovery dataset. We employed default values for all parameters and provided default reference data downloaded from https://resources.aertslab.org/cistarget/: Database (hg38_500bp_up_100bp_down_full_tx_v10_clust.genes_vs_motifs.rankings, hg38_10kbp_up_10kbp_down_full_tx_v10_clust.genes_vs_motifs.rankings), table (motifs-v10nr_clust-nr.hgnc-m0.001-o0.0.tbl), and transcription factor list (allTFs_hg38.txt). An additional TF list was downloaded from http://humantfs.ccbr.utoronto.ca/download/v_1.01/TF_names_v_1.01.txt and the two lists were merged totaling 2093 TFs.

The same resource files and parameters were used in the ROSMAP cohort replication, employing 7938 out of the 8561 genes used in the discovery cohort, including 894 out of the 896 TFs. Each regulon is represented by a single TF. We tested for an overall concordance in regulon-TFs between the discovery and ROSMAP datasets using the hypergeometric test implemented in the scipy.stats.hypergeom.pmf function (version 1.6.3), using M = 896 total TFs, *n* = 222 TFs in discovery, *N* = 190 TFs in ROSMAP, k = 122 intersecting TFs. We intersected the list of regulons from the discovery and ROSMAP, and then selected those that were differentially expressed in Oligo-TFEB state (Supplementary Dataset 8). Finally, we selected those TFs whose regulons showed a significant overlap among the list of target genes, using the hypergeometric test using M = 8561 total genes used in analysis, *n* = the number of genes in discovery regulon, *N* = number of genes in ROSMAP regulon, k = the number of intersecting genes (Supplementary Dataset 19). We employed the Benjamini–Hochberg method (alpha = 0.05) to correct for multiple testing. We then kept the significant regulons that had >2 intersecting target genes. We only used the genes that were differentially expressed in Oligo-TFEB when visualizing the network.

## Data visualization browser

We developed the Single Nucleus Alzheimer disease RNAseq Explorer (SNARE) to host the single nucleus expression data using the cellxgene[105] platform. It can be publicly and freely accessed at http://web.hararilab.org/SNARE/. We include the UMAP representations for the full dataset (consists of all cell types) and the individual cell type subclustering.

## Characterization of loci identified in AD risk GWAS

A list of genes identified through AD GWAS was collected. We started with 89 genes from 38 different loci that were previously prioritized[13,61,69]. We then added in all 1240 genes from all the significant loci. This list was filtered by genes detected in our snRNA-seq dataset, leaving 530 genes (77 of the prioritized genes). The significant cell state DE results were queried for the genes within our curated lists and extracted by cell type. We did not include results from direct comparisons with cell states that contained less than 5% of that cell type's nuclei. We removed gene hits with a negative estimate for DE analyses that were a cell state against all others (e.g., mic.1 vs all other mic cell states). We calculated the log2 fold-change from the estimates provided by nebula by using this equation:

$$\log_2 \text{FC} = log_2(e^{\text{estimate}}) \qquad (13)$$

We selected the maximum log fold-change in our plots if a gene had multiple hits within a single cell type. For each gene in our gene list, we calculated the mean expression in each cell type.

## Replication of GWAS loci characterization

SnRNA-seq data from UCI MIND's ADRC was accessed from Synapse[17]. The data was loaded into a Seurat object and normalized using *SCTransform* with "nCount_RNA" and "nFeature_RNA" as regression variables. Using just the 77 prioritized genes from above, we ran differential expression analysis between cell states and between each cell state and all other cell states at once within each cell type using the cluster annotations provided with the data. As before, the nebula[91] package was used for linear mixed models on the SCTransformed expression counts with cluster and sex as fixed effects and sample as a random effect. Following the processes above, we calculated the maximum log2FC and the average expression of these genes in each cell type. Genes with a *p* value less than 0.05 were included in the maximum log2FC calculation.

Average expression was log10 transformed and Pearson correlations were calculated for all genes and cell types at once and for each cell type individually using *cor.test* in R (Supplementary Fig. 14). Maximum log2FC values were converted to binary format (0 = gene not differential expressed between cell states, 1 = differentially expressed) for comparison creating 2 × 2 tables that could be passed to fisher exact tests (*fisher.test* in R).

## snATAC-seq data processing

We used CellRanger (version 6.1.1) to process the publicly available snATAC-seq fastq files on Synapse from the UCI MIND ADRC[17]. Reads were mapped to the GRCh38 reference obtained from https://cf.10xgenomics.com/supp/cell-arc/refdata-cellranger-arc-GRCh38-2020-A-2.0.0.tar.gz. We first filtered the BAM file outputted by Cell Ranger using samtools view with flags -f 3 -F 4 -f 8 -F 256 -F 1024 -F 2048 -q 30. We then used a custom python script to subset the filtered BAM file into one pseudo-bulk BAM file for each cell type using the original cluster assignments provided by the authors. We used MACS[106] (version 2.2.7.1) with flags−nomodel --shift -100 --extsize 200 --keep-dup all --call-summits -B to call narrow peaks on each cell type BAM file. We removed any narrow peaks overlapping blacklisted regions in the genome (https://www.encodeproject.org/files/ENCFF356LFX). For each cell type, we generated a matrix of snATAC-seq read counts per barcode for each narrow peak, which was used as input to CICERO[107] (version 1.4.4) to calculate co-accessibility across regions.

## snATAC-seq co-accessibility with AD GWAS

We downloaded the genetic fine-mapping results from Supplementary Dataset 8[61]. We used column finemap_prob_nc to obtain the posterior probability of association (PPA) values for the primary (and secondary, when available) credible sets per locus. Using only variants with PPA

>0.01, we identified all snATAC-seq peaks co-accessible with these variants in a cell-type-specific manner with an absolute CICERO[107] score ≥0.001. In addition, we identified all transcription start sites (TSS) that overlapped a fine-mapped GWAS variant and a snATAC-seq narrow peak. We used pyGenomeTracks[108] (version 3.7) to visualize the results.

## Statistics and reproducibility

All statistical methods and tests used in this paper are described as appropriate in the figure legends, methods, supplementary or main text. All instances where data were excluded from the further analysis are detailed above in the quality control descriptions. No statistical method was used to determine the sample size, but the frequency of the genetic variants in the general population was considered when selecting the sample size. Additionally, power analyses were performed on the anticipated sample size and nuclei counts. We planned to sequence 10,000 nuclei for each of the 74 samples for a total of ~750,000 nuclei. We expected to remove 50% of these during QC, leaving ~375,000. We expected the largest cell type cluster to make up 60% of these, providing the power (0.95) to detect cluster proportion differences with an effect size of Cohen's $f^2 = 2.16 \times 10^{-4}$ according to the R package pwr (version 1.3.0). Predicting that the largest subcluster within this cell type would account for 50% of the nuclei, we calculated Cohen's $d = 0.03$ for the largest of all our analyses. We anticipated the smallest models to include ~1000 nuclei, with the smallest subcluster accounting for 20% of these nuclei. This provided power (0.8) to detect $f^2 = 0.03$ and $d = 0.44$. Samples were classified into experimental groups on the basis of neuropathological analysis and clinical data. Analyses were controlled for individual-level covariates, including age and sex. Laboratory staff were blinded to sample status during sample preparation. Investigators were not blinded to group allocation during data collection and/or analysis. Knowledge of group allocation was required to perform differential abundance analysis.

## Reporting summary

Further information on research design is available in the Nature Portfolio Reporting Summary linked to this article.

## Data availability

The single nucleus data from the Knight ADRC generated in this study have been deposited in the National Institute on Aging Genetics of Alzheimer's Disease Data Storage Site (NIAGADS) with accession number NG00108. The raw single nucleus data from the DIAN brain bank generated in this study are available under restricted access to maintain individual and family confidentiality. These samples contain rare disease-causing variants that could be used to identify the participating individuals and families. Access can be obtained by request through the online resource request system on the DIAN Website: https://dian.wustl.edu/our-research/for-investigators/dian-observational-study-investigator-resources/. As detailed on the website, "Data requests will be reviewed based on the following criteria: (1) Scientific merit and feasibility (e.g., availability of DIAN resources to fulfill the request), (2) appropriateness of the investigator's qualifications and resources to protect the data, (3) appropriateness to DIAN goals/themes." Additional conditions of access include: "(1) the recipient to cite/reference the grant (Dominantly Inherited Alzheimer Network, U19AG032438) in any presentation or publication that may result from the research, (2) Should publications result from the use of DIAN resources now or in the future, the recipient agrees to notify the DIAN Executive Director with details (reference or PubMedCentral ID#) and provide a copy of the publication so productivity derived from [DIAN] resources can be reported to the funding agency (the National Institute on Aging (NIA)). Such publications require compliance with NIH public access policies and DIAN data sharing/publication policies, (3) Should funding result from this research now or in the future, please

notify the DIAN Executive Director with details (grant title, sponsor, number, dollar total, dates) so productivity derived from [DIAN] resources can be reported to NIA, (4) no sharing of data with a third party is allowed without the permission of the DIAN Steering Committee, (5) de-identified DIAN data will be made available to investigators to conduct analyses after approval by the PI and the relevant DIAN Core Leader. Allow 30–60 days for the review process and 30 days for interaction with the Biostatistics Core to provide the dataset." The processed single nucleus RNAseq data generated in this study can be freely viewed at http://web.hararilab.org/SNARE/. The 5xFAD mouse microglia data used in this study are in the Gene Expression Omnibus (GEO database) under the accession number GSE141917. The ROSMAP single nucleus RNA-sequencing data used in this study are available at Synapse under Synapse ID syn21125841. The single-nucleus RNA-sequencing and single-nucleus ATAC sequencing data used in this study from the UCI MIND ADRC are available at Synapse under Synapse ID syn22079621. The GRCH38 reference data used with CellRanger was downloaded here [ftp://ftp.ensembl.org/pub/release-93/fasta/homo_sapiens/dna/Homo_sapiens.GRCh38.dna.primary_assembly.fa.gz], [ftp://ftp.ensembl.org/pub/release-93/gtf/homo_sapiens/Homo_sapiens.GRCh38.93.gtf.gz]. The pySCENIC default reference data was downloaded from https://resources.aertslab.org/cistarget/: Database [https://resources.aertslab.org/cistarget/databases/homo_sapiens/hg38/refseq_r80/mc_v10_clust/gene_based/hg38_500bp_up_100bp_down_full_tx_v10_clust.genes_vs_motifs.rankings.feather], [https://resources.aertslab.org/cistarget/databases/homo_sapiens/hg38/refseq_r80/mc_v10_clust/gene_based/hg38_10kbp_up_10kbp_down_full_tx_v10_clust.genes_vs_motifs.rankings.feather], table [https://resources.aertslab.org/cistarget/motif2tf/motifs-v10nr_clust-nr.hgnc-m0.001-o0.0.tbl], and transcription factor list [https://resources.aertslab.org/cistarget/tf_lists/allTFs_hg38.txt]. An additional TF list was downloaded from [http://humantfs.ccbr.utoronto.ca/download/v_1.01/TF_names_v_1.01.txt]. The GRCH38 referenced used in the chromatic accessibility analysis was downloaded here [https://cf.10xgenomics.com/supp/cell-arc/refdata-cellranger-arc-GRCh38-2020-A-2.0.0.tar.gz]. And the blacklisted regions in the genome were downloaded here [https://www.encodeproject.org/files/ENCFF356LFX]. Source data are provided with this paper.

## Code availability

Custom code used to analyze the snRNA-seq data and datasets generated and/or analyzed in the current study are available from the corresponding authors upon request or at https://github.com/HarariLab/parietal-snRNAseq[89].

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

## Acknowledgements

Doctors Celeste M. Karch, Bruno A. Benitez, and Oscar Harari con-
tributed equally to this work as co-senior authors. Data collection and
sharing for this project were supported by The Dominantly Inherited
Alzheimer Network (DIAN, U19AG032438), funded by the National
Institute on Aging (NIA), the Alzheimer's Association (SG-20-690363-
DIAN), the German Center for Neurodegenerative Diseases (DZNE), Raul
Carrea Institute for Neurological Research (FLENI), Partial support by the
Research and Development Grants for Dementia from Japan Agency for
Medical Research and Development, AMED, and the Korea Health
Technology R&D Project through the Korea Health Industry Develop-
ment Institute (KHIDI), Spanish Institute of Health Carlos III (ISCIII),
Canadian Institutes of Health Research (CIHR), Canadian Consortium of
Neurodegeneration and Aging, Brain Canada Foundation, and Fonds de
Recherche du Québec – Santé. DIAN Study investigators have reviewed
this manuscript for scientific content and consistency of data inter-
pretation with previous DIAN Study publications. We acknowledge the
altruism of the participants and their families and the contributions of the
DIAN research and support staff at each of the participating sites for their
contributions to this study. The results published here are partly based
on data obtained from the AD Knowledge Portal (https://
adknowledgeportal.org). Study data were provided by the Rush Alz-
heimer's Disease Center, Rush University Medical Center, Chicago. Data
collection was supported through funding by NIA grants P30AG10161
(ROS), R01AG15819 (ROSMAP; genomics and RNAseq), R01AG17917
(MAP), R01AG30146, R01AG36042 (5hC methylation, ATAC-seq),
RC2AG036547 (H3K9Ac), R01AG36836 (RNAseq), R01AG48015
(monocyte RNAseq) RF1AG57473 (single nucleus RNAseq),
U01AG32984 (genomic and whole exome sequencing), U01AG46152
(ROSMAP AMP-AD, targeted proteomics), U01AG46161(TMT pro-
teomics), U01AG61356 (whole genome sequencing, targeted pro-
teomics, ROSMAP AMP-AD), the Illinois Department of Public Health
(ROSMAP), and the Translational Genomics Research Institute (geno-
mic). Additional phenotypic data can be requested at www.radc.rush.
edu. This work was possible thanks to the following governmental grants
from the National Institute of Health: NIA R01AG057777 (O.H.),
R56AG067764 (O.H.), U01AG072464 (O.H. and C.M.K.), R01AG074012
(O.H.) P30AG066444 (JCM), P01AGO26276 (JCM), U19AG032438 (RJB),
P01AG003991 (J.C.M.), NINDS R01NS118146 (B.A.B.), R21NS127211
(B.A.B.), RF1AG071706 (B.A.B.), RFNS110809 (C.M.K.), R01AG062734
(C.M.K.), NIA T32AG058518 (L.B.), the BrightFocus Foundation (C.M.K.),
and the Chan Zuckerberg Initiative (C.M.K.). O.H. is an Archer Foundation
Research Scientist. This work was supported by access to equipment
made possible by the Hope Center for Neurological Disorders and the
Departments of Neurology and Psychiatry at Washington University
School of Medicine. The funders of the study had no role in the collec-
tion, analysis, or interpretation of data, in the writing of the report, or in
the decision to submit the paper for publication. Fig 1a was created with
BioRender.com

## Author contributions

L.B. performed the quality control, analysis, interpreted the results,
created the figures, and wrote the manuscript. S.-F.Y. performed the DE
analyses on the Oligodendrocytes and interpreted the results for the
APOE DE analyses. J.L.D.-A. performed initial quality control and cell
clustering and participated in writing the introduction and methods
sections. Y.D. performed the subclustering and DE analyses on the
Astrocytes and participated in interpreting the Astrocytes' DE results.
B.C.N. participated in data analysis. C.S.-T. participated in data quality
control. T.D. performed mouse scRNA-seq analysis and integration with
human snRNA-seq, and edited the manuscript. R.D.A. performed the
snATAC-seq co-accessibility analysis and interpretation and participated
in figure creation and manuscript review. M.V.F, J.P.B., and K.B. were
involved in sample processing and data management. J.C.M., R.J.B.,
R.J.P., E.M., C.X., A.G., M.F., were involved in the sample collection,
profiling, and provided a critical review of the manuscript. G.T.S. parti-
cipated in data interpretation and critical review of the manuscript. J.K.
provided oversight of the mouse scRNA-seq and integration. B.A.B.
optimized protocol for nuclei extractions and supervised single-cell data
generation. C.M.K. and B.A.B. were involved in experiment design,
intellectual contribution, and manuscript writing and review. O.H.
supervised all experiment design, analyses, and data interpretation and
edited the manuscript.

## Competing interests

J.C.M. is a consultant for the Barcelona Brain Research Center (BBRC)
and the TS Srinivasan Advisory Board. J.C.M. is an advisory board
member for the Cure Alzheimer's Fund Research Strategy Council. R.J.B.
maintains an equity ownership interest and is a member of the advisory
board of C2N Diagnostics. Unrelated to this article, R.J.B. serves as the
principal investigator of the DIAN-TU, which the Alzheimer's Association
supports, GHR Foundation, an anonymous organization, and the DIAN-
TU Pharma Consortium (Active: Eli Lilly and Company/Avid Radio-
pharmaceuticals, F. Hoffman-La Roche/Genentech, Biogen, Eisai, and
Janssen. Previous: Abbvie, Amgen, AstraZeneca, Forum, Mithridion,
Novartis, Pfizer, Sanofi, and United Neuroscience). In addition, in-kind
support has been received from CogState and Signant Health. Unrelated
to this article, R.J.B. has submitted the US nonprovisional patent appli-
cation "Methods for Measuring the Metabolism of CNS Derived Biomo-
lecules in Vivo" and provisional patent application "Plasma Based
Methods for Detecting CNS Amyloid Deposition." E.M. receives research

support from the NIA, Hoffman-La Roche, and Eli Lilly, is a member of advisory boards for Eli Lilly, Alector, and the NIA, and holds a leadership role in Fondation Alzheimer and Alzamend. C.X. is a consultant for DIADEM and a member of the advisory board for the University of Wisconsin ADRC. A.G. receives royalties from Athena Diagnostics and Taconic Biosciences, is a consultant for Genentech SAB and AbbVie and holds stock or stock options in Cognition Therapeutics and Denali Therapeutics. M.F. receives research support from Eli Lilly and Company, Hoffmann-La Roche, Avanir, Biogen, Cognition Therapies, Green Valley, Otsuka, Neurotrope Biosciences, AZTherapies, Athira, Ionis, and Lexeo, and is a member of advisory boards for Oligomerik and T3D. The remaining authors declare no competing interests.

## Additional information

[1]Department of Psychiatry, Washington University School of Medicine in St. Louis, St. Louis, MO, USA. [2]Hope Center for Neurological Disorders, Washington University School of Medicine in St. Louis, St. Louis, MO, USA. [3]NeuroGenomics and Informatics, Department of Psychiatry, Washington University School of Medicine in St. Louis, St. Louis, MO, USA. [4]Merck & Co., Inc., Boston, MA, USA. [5]Baylor College of Medicine, Houston, TX, USA. [6]Department of Pathology and Immunology, Washington University School of Medicine in St. Louis, St. Louis, MO, USA. [7]Center for Brain Immunology and Glia (BIG), Washington University School of Medicine in St. Louis, St. Louis, MO, USA. [8]Knight Alzheimer Disease Research Center, Washington University School of Medicine in St. Louis, St. Louis, MO, USA. [9]Department of Neurology, Washington University School of Medicine in St. Louis, St. Louis, MO, USA. [10]Division of Biostatistics, Washington University School of Medicine in St. Louis, St. Louis, MO, USA. [11]Ronald M. Loeb Center for Alzheimer's Disease, Department of Genetics and Genomic Sciences, Icahn School of Medicine at Mount Sinai, New York, NY, USA. [12]Department of Neurology, Indiana University School of Medicine, Indianapolis, IN, USA. [13]School of Medical Sciences and Charles Perkins Centre, Faculty of Medicine and Health, The University of Sydney, Sydney, NSW, Australia. [14]Department of Neurology, Beth Israel Deaconess Medical Center, Harvard Medical School, Boston, MA, USA. [15]These authors contributed equally: Celeste M. Karch, Bruno A. Benitez, Oscar Harari. ✉e-mail: harario@wustl.edu

## Dominantly Inherited Alzheimer Network (DIAN)

**Maria Victoria Fernandez** [1,2,3], **Eric McDade** [1], **Celeste M. Karch** [1,2,3,15], **Richard J. Perrin** [2,6,8,9], **Randall J. Bateman** [2,8,9], **Chengjie Xiong** [8,10], **Alison M. Goate** [11] **& Martin Farlow** [12]

