## [Peer Review File · Nature Communications]

Single-nucleus RNA-sequencing of autosomal dominant Alzheimer disease and risk variant carriersEditorial Note: This manuscript has been previously reviewed at another journal that is not operating a transparent peer review scheme. This document only contains reviewer comments and rebuttal letters for versions considered at *Nature Communications*.

REVIEWER COMMENTS

Reviewer #1 (Remarks to the Author):

In the revised version, the authors have added several analysis, but still fall short of 2 important things - data quality and limited transcriptomic analysis. In this regard, this reviewer proposes these revisions -

a) Sample level QC violinplots, showing UMI, number of genes, percent mt-reads. Also same QC post-filtering for each cluster. Please include violinplots and not tables. Much of the data is in Supplemental tables, and the authors should make efforts to visualize them.

b) Perform DE analysis using pseudotime, authors can use monocle, paga , softwares. and plot DE genes for each cell-type using pseudotime

c) Perform gene-regulatory network analysis using SCENIC

d) Perform cell-signaling analysis using packages like cellChat.

Without these analysis, much of the data is hidden in supplemental tables, and the paper in its current form very weak.

Reviewer #2 (Remarks to the Author):

Reviewer #2:

The authors responded to most of the previous queries. However, given the current study's focus on understanding the effects of rare coding mutations in AD, the small number of samples harboring the mutations that was examined in this study remains to be a major concern, as the conclusion can be affected by variations in sample quality. Thus, the authors should perform additional quality controls, and exclude or redo the sequencing of samples that showed abnormal results in the current datasets (including the number of droplets and the fraction of cell types).

The following information should also be included in the revised manuscript:

1. Given that the authors have used TREM2 (R47H, R62H, H157Y); R46H APOE (2, 3, 4); and MS4A (rs1582763) genotypes for the analysis, they should explicitly list the genotypes of each variant in Supplementary Table 1.

2. As ethnic information was used in the analysis, the authors should provide the ethnic information for each donor accordingly.

3. The term "Three pairs of samples" is confusing. Are those pairs from the same family and/or in relationship with each other? If so, consider using the term "family" to group those pairs, and provide the exact relationships among those family members. Again, as this information was used in the analysis, the authors should provide the relevant information accordingly (e.g., adding a new column in Supplementary Table 1 with this information).

In addition, the authors should provide a label to indicate the color scale for the heatmap in Figure 2b.

Reviewer #3

The authors have provided compelling data, but my major concern still lies in the variation among the samples.

Technically speaking, the number of droplets should follow a normal distribution if the same protocol is applied to the samples under the same conditions. For the 67 analyzed samples (Supplementary Table 2), the average number of nuclei was 4,389 and the standard deviation was 2,920. However, for the samples retained for analysis, the number of droplets varied between 314 and 15,530, which does not seem likely to arise normally. The authors should demonstrate if those variations indeed represent true biological variations. Specifically, they should apply rigorous filtering to exclude any outliers resulting from issues during sample preparation or library construction. It is important to determine that the outliers are not the mutation carriers contributed to the observed variations.

Some of the data (Supplementary Table 4) raises further concerns:

- Only a few samples contributed predominantly to specific subclusters of cells (e.g., Astro.2, Astro.4, Micro.4, Micro.8, Oligo.5, Oligo.8).**
- Three of the 67 Oligo.5 samples (sample42, sample55, and sample58) contributed more than 87% of the Oligo.5 nuclei (the cluster of cells discussed in Figure 4).**
- Eight out of 67 samples detected aberrant neuron nuclei (Supplementary Table 4).**

These observations raise substantial concerns about the quality of the sample data. In particular, since most of the results focus on rare coding mutations found in a small number of samples, the inclusion of some low-quality samples might strongly affect the conclusion. If most of the outliers were not mutation carriers, removing them from the analysis would not affect the main conclusion of the study.

REVIEWER COMMENTS

We thank reviewers #1 and #2 for their contribution to the review process and comments. Below we provide detail responses raised in this second round of revision:

Reviewer #1 (Remarks to the Author):

In the revised version, the authors have added several analysis, but still fall short of 2 important things - data quality and limited transcriptomic analysis. In this regard, this reviewer proposes these revisions -

a) Sample level QC violinplots, showing UMI, number of genes, percent mt-reads. Also same QC post-filtering for each cluster. Please include violinplots and not tables. Much of the data is in Supplemental tables, and the authors should make efforts to visualize them.

We thank the reviewer for their suggestion. We have added these plots to the supplementary figures (Supplementary Figure 16, 17). The plots below are linked to online version.

Supplementary Figure 16 QC metrics by sample.

Violin plots of UMI counts, gene counts and percent of mitochondrial reads for each sample.

Supplementary Figure 17 QC metrics by cell state.

Violin plots of UMI counts, gene counts and percent of mitochondrial reads for each cell state.

b) Perform DE analysis using pseudotime, authors can use monocle, paga , softwares. and plot DE genes for each cell-type using pseudotime

We thank the reviewer for showing such great interest in our data and its vast potential. We are certainly interested in using pseudotime to interrogate this data. We believe that this approach might be highly informative to understand the transition between cellular states, and highly important, to determine how the proximity to amyloid plaques and other neuropathological insults modulate the transcriptional states. We have leveraged these approaches in our previous publications, but at the present time, and after a lot of internal debate and exploration, we consider that these approaches should be coupled with single-cell CRISPR screens and/or spatial transcriptomics plus immunohistochemistry staining. We believe we do not have the additional key data to replicate or validate any results of the pseudotime analyses in a meaningful way.

c) Perform gene-regulatory network analysis using SCENIC

We thank the reviewer for this suggestion. We ran pySCENIC on the Oligo-TFEB cell state and added those results to the appropriate results and methods sections.

“Furthermore, our analysis of the expression-derived gene regulatory network (GRN) for these oligodendrocytes identified transcription factors mediating regulation in both discovery and replication data (replication $P=9.08 \times 10^{-41}$). We identified *SOX8*, *SREBF1*, and *NKX6-2* linked to myelination^{17,52,53}, *NFE2L2/NRF2* associated with multiple AD pathologies including A β , Tau, and oxidative stress⁵⁴, and *ZNF518A* associated with increased somatic-mutational burden in AD oligodendrocytes⁵⁵ (Figure 4g, Supplementary Table 19). Downregulation of *SREBF1* was

previously reported in oligodendrocytes in AD brain tissue¹⁷, and associated with regulation of *ErbB/mTOR* signaling pathways and autophagy^{56,57}.”

“Reconstruction of gene regulatory network from snRNA-seq

We employed the python implementation of the SCENIC¹⁰¹ analysis method called pySCENIC¹⁰² to evaluate the gene regulatory networks (GRN). We selected 8,561 genes expressed in $\geq 5\%$ of the full oligodendrocyte population in the discovery dataset. We employed default values for all parameters and provided default reference data downloaded from <https://resources.aertslab.org/cistarget/>: Database (hg38_500bp_up_100bp_down_full_tx_v10_clust.genes_vs_motifs.rankings, hg38_10kbp_up_10kbp_down_full_tx_v10_clust.genes_vs_motifs.rankings), table (motifs_v10nr_clust-nr.hgnc-m0.001-o0.0.tbl), and transcription factor list (allTFs_hg38.txt). An additional TF list was downloaded from http://humantfs.ccb.utoronto.ca/download/v_1.01/TF_names_v_1.01.txt and the two lists were merged totaling in 2,093 TFs.

The same resource files and parameters were used in the ROSMAP cohort replication, employing 7,938 out of the 8,561 genes used in the discovery cohort including 894 out of the 896 TFs. Each regulon is represented by a single TF. We tested for an overall concordance in regulon-TFs between the discovery and ROSMAP data sets using the hypergeometric test implemented in the `scipy.stats.hypergeom.pmf` function (version 1.6.3), using $M=896$ total TFs, $n=222$ TFs in discovery, $N=190$ TFs in ROSMAP, $k=122$ intersecting TFs. We intersected the list of regulons from the discovery and ROSMAP, and then selected those that were differentially expressed in Oligo-TFEB state (Supplementary Table 8). Finally, we selected those TFs whose regulons showed a significant overlap among the list of target genes, using the hypergeometric test using $M=8,561$ total genes used in analysis, $n=$ number of genes in discovery regulon, $N=$ number of genes in ROSMAP regulon, $k=$ number of intersecting genes (Supplementary Table 19). We employed the Benjamini-Hochberg method ($\alpha = 0.05$) to correct for multiple testing. We then kept the significant regulons that had >2 intersecting target genes. We only used the genes that were differentially expressed in Oligo-TFEB when visualizing the network.”

Figure 4g,h Oligo-TFEB GRN

g) Identification of gene regulatory networks (GRN) in Oligo-TFEB discovery and replication cohorts. Regulons were filtered to include only those identified in both cohorts ($p=9.98 \times 10^{-41}$) with significant differential expression in Oligo-TFEB. Then those regulons with significant ($\text{Adj.P} < 0.05$) coincidence in the underlying target genes between cohorts were selected. h) Gene regulatory network for transcription factors (TF; shown in blue) replicated in both discovery (purple edges) and ROSMAP (orange edges) datasets for oligo-TFEB. Replicated target genes are shown in yellow and edges in green. Genes within AD GWAS loci are highlighted in red. g,h) (*) = $\text{Adj.P} < 0.05$, (**) = $\text{Adj.P} < 0.01$.

d) Perform cell-signaling analysis using packages like cellChat.

We thank the reviewer for suggesting an analysis on cell-cell communication. We also believe this is a highly important and informative approach to interrogate the data. Indeed, we have been working on cross-talk approaches, and the amount of information and insights learned deserve its own publication. Otherwise, many of the insights would be missed in a single crowded publication. Please refer to our preprint, "Systematic characterization of brain cellular crosstalk signaling networks in Alzheimer's disease reveals a novel role for SEMA6D in TREM2-dependent microglial activation".

Without these analysis, much of the data is hidden in supplemental tables, and the paper in its current form very weak.

We are also excited by this data and feel it should be interrogated in many additional ways to study new aspects in ADAD and sporadic AD. As is the nature of most published work, there is only room for interesting stories and detailed highlights following a single thread of reasoning. We believe that as currently shaped, this manuscript already provides many impactful results. It would be detrimental to squeeze in additional analyses, burying results we deem worthy of highlighting.

Reviewer #2 (Remarks to the Author):

Reviewer #2:

The authors responded to most of the previous queries. However, given the current study's focus on understanding the effects of rare coding mutations in AD, the small number of samples harboring the mutations that was examined in this study remains to be a major concern, as the conclusion can be affected by variations in sample quality. Thus, the authors should perform additional quality controls, and exclude or redo the sequencing of samples that showed abnormal results in the current datasets (including the number of droplets and the fraction of cell types).

We thank the reviewer for expressing this concern and are sorry for the confusion. For each of our cell type specific analyses, we did exclude samples that did not have sufficient nuclei counts ($n=60$) for that cell type (Supplementary Table 4,9). Additionally, the differential expression analyses were not conducted using the default functions in Seurat but using mixed models that account for the sample of origin for each nucleus (Nebula R package). Similarly, fraction of cell-types is also controlled in the analytical models. In addition, we want to mention that our data cleaning and quality control procedure evaluated data at a batch, donor and cell-type level, and discarded 7 samples, including one carrier of mutation in *PSEN1*. Unfortunately, the tissue from many of these donors is sparse and not available for re-sequencing. Further discussion on the cell counts for each sample can be found below in comment 1 to reviewer 3.

We want to emphasize that this is the first snRNA-seq data for ADAD samples and one of the first snRNA-seq studies designed to address the genetic factors in ADAD and sporadic AD. We have sequenced 9 neuropathological controls, 16 ADAD, 16 sporadic AD, and 19 TREM2 (13 reduced activity) samples. Not only do these sample sizes compete with published analyses, but they frequently outnumber them^{2,7-9}. Given the low frequency of the ADAD mutation and TREM2 variants, we grouped these donors. In the updated manuscript, we recognize this limitation in the discussion.

“These analyses are limited by the rarity of mutations in *APP*, *PSEN1*, and low frequency of variants in *TREM2* in the general population. As a result, samples with mutations in *APP* and *PSEN1* were merged and considered as a single ADAD group and three *TREM2* variants (p.R47H, p.R62H, and p.H157Y) were merged as the *TREM2*-reduced-activation group despite there being slight

differences in the functional mechanisms. Further analyses on additional tissue samples from carriers of these variants are needed to fully uncover the variant specific effects in these critical AD genes.”

The following information should also be included in the revised manuscript:

1. Given that the authors have used TREM2 (R47H, R62H, H157Y); R46H APOE (2, 3, 4); and MS4A (rs1582763) genotypes for the analysis, they should explicitly list the genotypes of each variant in Supplementary Table 1.

We thank the reviewer for requesting this genetic information. We have added the APOE genotypes for each sample to in Supplementary Table 1. The MS4A genotypes were already present. The rarity of the TREM2 variants prohibits the sharing of exact sample mutations because it increases the chance for patient identification. However, the samples with TREM2 and TREM2 reduced activation variants are shown in Supplementary Table 1. Exact variants for TREM2, APP, and PSEN2 are available through NIAGADs and DIAN.

2. As ethnic information was used in the analysis, the authors should provide the ethnic information for each donor accordingly.

We thank the reviewer for requesting this information. Because of the low number of non-European brain donors included in our study and in the Knight ADRC and DIAN biobanks, we cannot share that information in the manuscript, as these samples and their families can be more easily identified. Nevertheless, ethnic information is accessible through NIAGADs and DIAN sharing protocols.

3. The term “Three pairs of samples” is confusing. Are those pairs from the same family and/or in relationship with each other? If so, consider using the term “family” to group those pairs, and provide the exact relationships among those family members. Again, as this information was used in the analysis, the authors should provide the relevant information accordingly (e.g., adding a new column in Supplementary Table 1 with this information).

We thank the review for bringing this confusion to our attention. We have changed the wording in the manuscript and added exact relationships to the table (Supplementary Table 26) for the Knight ADRC donors. We cannot share additional familial relationship for DIAN donors, as it could be identifying. This data is available upon request to the Dominantly Inherited Alzheimer Network.

“The data generated is primarily from unrelated donors, but also includes data from related samples from three nuclear families. In more details, two pairs from two sibships from the Knight ADRC, and one pair of related donors from the DIAN cohort (sAD-family, sAD-presymptomatic family, and ADAD-family).”

In addition, the authors should provide a label to indicate the color scale for the heatmap in Figure 2b.

We thank the reviewer for this suggestion. We have added the missing label to Figure 2b as can be seen below.

Figure 2b The color scale has been labelled and unified across cell types.

Reviewer #3

The authors have provided compelling data, but my major concern still lies in the variation among the samples.

Technically speaking, the number of droplets should follow a normal distribution if the same protocol is applied to the samples under the same conditions. For the 67 analyzed samples (Supplementary Table 2), the average number of nuclei was 4,389 and the standard deviation was 2,920. However, for the samples retained for analysis, the number of droplets varied between 314 and 15,530, which does not seem likely to arise normally. The authors should demonstrate if those variations indeed represent true biological variations. Specifically, they should apply rigorous filtering to exclude any outliers resulting from issues during sample preparation or library construction. It is important to determine that the outliers are not the mutation carriers contributed to the observed variations.

We thank the reviewer for seeking to confirm the integrity and quality of our data. We have compared our nuclei counts by sample to those from the literature and our study is consistent with those already published^{2,7-13} (Rebuttal Figure 1). Additionally, the one outlier observed in our data is a rare ADAD individual, sample46, with limited tissue availability. We reviewed the distributions for number of UMIs, number of genes, and percent of mitochondrial reads for sample46 and none of those QC metrics indicate poor quality (Supplementary Figure 16).

Rebuttal Figure 1 This study's cell counts by sample distribution is consistent with those published in the literature. Counts were scaled by study for accurate comparison across studies. A single sample, sample46, from this study is considered an outlier in nuclei counts but does not show poor quality.

Some of the data (Supplementary Table 4) raises further concerns:

- Only a few samples contributed predominantly to specific subclusters of cells (e.g., Astro.2, Astro.4, Micro.4, Micro.8, Oligo.5, Oligo.8).

We thank the reviewer for raising this concern. We would like to emphasize that some of these clusters have a very low number of nuclei, and we do not claim any specific biological interpretation based on these. For example, Micro.8 and Oligo.8 are the smallest clusters for their respective cell types and were not specifically discussed. Furthermore, the remaining clusters highlighted by the reviewer are associated with TREM2 reduced activation carriers (Oligo.5) or ADAD samples (Astro.2, Astro.4, Micro.4), which explains their specificity. It is worth mentioning that our analytical approach corrects for biases and artifacts by employing mixed models which groups nuclei by donor, among other covariates. Furthermore, we would like to mention that we employed data from additional cohorts and sources, including ROSMAP and 5xFAD mouse models respectively to replicate the findings.

- Three of the 67 Oligo.5 samples (sample42, sample55, and sample58) contributed more than 87% of the Oligo.5 nuclei (the cluster of cells discussed in Figure 4).

We thank the reviewer for highlighting this point. The Oligo.5 (Oligo-TFEB) expression signature and its association with TREM2 variant carriers were replicated in an independent ROSMAP cohort (Supplementary Figure 9, 11). Additionally, the differential expression analyses were run using Nebula a tool that implements mixed effects models that account for the sample of origin for each nucleus. As shown in the table above, this cell state had many samples contributing to the cluster even if their contributions were smaller than those from sample42, sample55, and sample58. We also investigated the number of nuclei, UMIs, and genes for samples 42, 55, and 58. None of them are outliers as can be seen in Supplementary Figure 16 and Rebuttal Figure 1.

- Eight out of 67 samples detected aberrant neuron nuclei (Supplementary Table 4).

We thank the reviewer for requesting clarification. These eight samples were not included in any of the neuronal analyses including the proportion and differential expression analyses. A detailed breakdown of the number of nuclei and number of samples used in each regression analysis is found in Supplementary Table 9.

We also investigated these eight samples with low neuron counts in the other cell types to confirm there were no artifacts. We observed similar cell-type-specific distributions for these eight samples as we did in all other samples for UMI counts, gene counts, percent mitochondrial reads, and nuclei counts as can be seen in Rebuttal Figure 2.

Rebuttal Figure 2 Concordance between the eight neuron low count samples and all other samples according to QC measures. a-d) The eight low-neuron-count samples are grouped together in blue. All other samples are grouped together in red. a) Median UMI counts by sample. b) Median gene counts by sample. c) Median percent mitochondrial reads by sample. d) The log transformed nuclei counts for each cell type by sample.

These observations raise substantial concerns about the quality of the sample data. In particular, since most of the results focus on rare coding mutations found in a small number of samples, the inclusion of some low-quality samples might strongly affect the conclusion. If most of the outliers were not mutation carriers, removing them from the analysis would not affect the main conclusion of the study.

We thank the reviewer for their time and attention while reading our manuscript. As mentioned above, this is the first snRNA-seq data for ADAD samples and one of the first snRNA-seq studies that attempts to directly address the genetic factors influencing AD. Additionally, we have 9 neuropathological controls, 16 ADAD, 16 sporadic AD, and 19 TREM2 (13 TREM2 reduced) AD samples. Not only do these sample sizes compete with published analyses, but they frequently out number them^{2,7-9}. We also want to reclarify that samples that did not pass the threshold for containing enough nuclei in the cell type were excluded from our analyses. Furthermore, we accounted for sample variability using mixed models and replicated significant associations in independent data sets.

References

- 1 Turnescu, T. *et al.* Sox8 and Sox10 jointly maintain myelin gene expression in oligodendrocytes. *Glia* **66**, 279-294 (2018). <https://doi.org:10.1002/glia.23242>
- 2 Morabito, S. *et al.* Single-nucleus chromatin accessibility and transcriptomic characterization of Alzheimer's disease. *Nat Genet* **53**, 1143-1155 (2021). <https://doi.org:10.1038/s41588-021-00894-z>
- 3 Chelban, V. *et al.* Mutations in NKX6-2 Cause Progressive Spastic Ataxia and Hypomyelination. *Am J Hum Genet* **100**, 969-977 (2017). <https://doi.org:10.1016/j.ajhg.2017.05.009>
- 4 Saha, S., Buttari, B., Profumo, E., Tucci, P. & Saso, L. A Perspective on Nrf2 Signaling Pathway for Neuroinflammation: A Potential Therapeutic Target in Alzheimer's and Parkinson's Diseases. *Front Cell Neurosci* **15**, 787258 (2021). <https://doi.org:10.3389/fncel.2021.787258>
- 5 Aibar, S. *et al.* SCENIC: single-cell regulatory network inference and clustering. *Nat Methods* **14**, 1083-1086 (2017). <https://doi.org:10.1038/nmeth.4463>
- 6 Kumar, N., Mishra, B., Athar, M. & Mukhtar, S. Inference of Gene Regulatory Network from Single-Cell Transcriptomic Data Using pySCENIC. *Methods Mol Biol* **2328**, 171-182 (2021). https://doi.org:10.1007/978-1-0716-1534-8_10
- 7 Zhou, Y. *et al.* Human and mouse single-nucleus transcriptomics reveal TREM2-dependent and TREM2-independent cellular responses in Alzheimer's disease. *Nat Med* **26**, 131-142 (2020). <https://doi.org:10.1038/s41591-019-0695-9>
- 8 Lau, S. F., Cao, H., Fu, A. K. Y. & Ip, N. Y. Single-nucleus transcriptome analysis reveals dysregulation of angiogenic endothelial cells and neuroprotective glia in Alzheimer's disease. *Proc Natl Acad Sci U S A* **117**, 25800-25809 (2020). <https://doi.org:10.1073/pnas.2008762117>
- 9 Yang, A. C. *et al.* A human brain vascular atlas reveals diverse mediators of Alzheimer's risk. *Nature* **603**, 885-892 (2022). <https://doi.org:10.1038/s41586-021-04369-3>
- 10 Zalocusky, K. A. *et al.* Neuronal ApoE upregulates MHC-I expression to drive selective neurodegeneration in Alzheimer's disease. *Nat Neurosci* **24**, 786-798 (2021). <https://doi.org:10.1038/s41593-021-00851-3>
- 11 Griswold, A. J. *et al.* Increased APOE ϵ 4 expression is associated with the difference in Alzheimer's disease risk from diverse ancestral backgrounds. *Alzheimers Dement* **17**, 1179-1188 (2021). <https://doi.org:10.1002/alz.12287>
- 12 Olah, M. *et al.* Single cell RNA sequencing of human microglia uncovers a subset associated with Alzheimer's disease. *Nat Commun* **11**, 6129 (2020). <https://doi.org:10.1038/s41467-020-19737-2>
- 13 Garcia, F. J. *et al.* Single-cell dissection of the human brain vasculature. *Nature* **603**, 893-899 (2022). <https://doi.org:10.1038/s41586-022-04521-7>

REVIEWERS' COMMENTS

Reviewer #1 (Remarks to the Author):

The authors have addressed all issues and the manuscript should be accepted for publication

Reviewer #2 (Remarks to the Author):

The authors have adequately addressed the concerns on sample quality, and included sufficient details of the QC process in the revised version. The reviewer agrees that this is the first scRNA-seq paper reporting the effect of genetic variants on AD brain. The current version is suitable to be published.

REVIEWERS' COMMENTS

Reviewer #1 (Remarks to the Author):

The authors have addressed all issues and the manuscript should be accepted for publication
We thank the reviewer for their time and energy contributed to the review process.

Reviewer #2 (Remarks to the Author):

The authors have adequately addressed the concerns on sample quality, and included sufficient details of the QC process in the revised version. The reviewer agrees that this is the first scRNA-seq paper reporting the effect of genetic variants on AD brain. The current version is suitable to be published.
We thank the reviewer for the detailed review of our manuscript and their contribution to the review process.